https://doi.org/10.1038/s41467-022-29742-2　　**OPEN**

# A computational theory of the subjective experience of flow

David E. Melnikoff [1✉], Ryan W. Carlson [2] & Paul E. Stillman[3]

Flow is a subjective state characterized by immersion and engagement in one's current activity. The benefits of flow for productivity and health are well-documented, but a rigorous description of the flow-generating process remains elusive. Here we develop and empirically test a theory of flow's computational substrates: the informational theory of flow. Our theory draws on the concept of mutual information, a fundamental quantity in information theory that quantifies the strength of association between two variables. We propose that the mutual information between desired end states and means of attaining them — $I(M;E)$ — gives rise to flow. We support our theory across five experiments (four preregistered) by showing, across multiple activities, that increasing $I(M;E)$ increases flow and has important downstream benefits, including enhanced attention and enjoyment. We rule out alternative constructs including alternative metrics of associative strength, psychological constructs previously shown to predict flow, and various forms of instrumental value.

[1] Department of Psychology, Northeastern University, Boston, MA, USA. [2] Department of Psychology, Yale University, New Haven, CT, USA. [3] Department of Marketing, Yale School of Management, Yale University, New Haven, CT, USA. ✉email: davidemelnikoff@gmail.com

Peak human performance emerges during the experience of flow—the subjective state of being immersed in one's current task[1–3]. Flow enhances learning and academic achievement[4–6], boosts productivity, fosters artistic talent[7], and improves objective measures of athletic skill[8,9]. Flow also promotes subjective well-being[7,10]; positive affect has been found to increase with the amount of flow experienced during the workday[11], and the absence of flow has been linked to depression[2].

Flow is a potential wellspring of achievement and well-being, but often goes underutilized. People frequently find their most important tasks tedious rather than immersive[1,2,10,12]. To help people reap the benefits of flow, we must achieve a deeper understanding of the mechanisms through which flow emerges. Indeed, existing descriptions of the flow-generating process are underspecified, cast in terms of abstract concepts rather than mathematically precise computational models. Only by grounding the flow-generating process in formal theoretical structures can we identify precisely which parameters must be adjusted, and by how much, to maximize flow for particular people and contexts. We pursue this aim in the present paper by proposing and testing a computational theory of flow.

Our proposal draws inspiration from the disparate fields of social psychology and artificial intelligence, which have converged on similar ideas with relevance to flow. Social psychologists have developed the concept of means-ends fusion to explain what makes some activities more intrinsically interesting than others[13–17]. The idea is that intrinsic interest emerges from mental associations between desired end states (e.g., bowling a strike) and means of attaining them (e.g., rolling a ball); the more a means and end are associated—or fused—the more interest the means evokes. The concept of intrinsic interest is, if not identical to flow, closely related[3], suggesting that the abstract notion of means-ends fusion could guide and constrain the search for flow-generating mechanisms.

Intriguingly, what could be interpreted as a formal definition of means-ends fusion appears in the field of artificial intelligence. It has proven useful to have artificial agents aim to maximize a quantity called empowerment: the maximum of the mutual information between the agent's actions and end states[18,19]. Mutual information is a fundamental quantity from information theory that measures the degree of association between two random variables[20]. Accordingly, the mutual information between actions and end states can be interpreted as the associative strength, or fusion, between means and ends. Other formulations of means-ends fusion are possible, but mutual information is an especially promising candidate, in part because empowerment-maximizing agents are reminiscent of humans experiencing flow: people tend to pursue flow-inducing activities for their own sake[3], and agents that maximize empowerment tend to learn, explore, and act meaningfully in the absence of external rewards and punishments[21–26].

The consilience between means-ends fusion and empowerment led us to integrate these concepts into a computational theory of flow. The crux of our proposal is this: Flow is an increasing function of the mutual information between desired end states and means of attaining them. As the mutual information between a means and its end increases, so does the degree of flow. We call this the informational theory of flow. Next, we specify what we mean by "means" and "ends," and how the mutual information between them is computed.

We equate means and ends with the random variables $M$ and $E$, respectively, where $M$ denotes a state brought about to achieve a goal, and $E$ denotes the outcome of goal-pursuit. Most activities can be represented in multiple ways, making $M$ and $E$ perceiver-dependent[27]. Consider a dart-throwing game that rewards players

for hitting a bullseye. For one person, $M$ and $E$ could be binary variables denoting whether the bullseye was hit or missed, and whether or not a reward was received, respectively. For someone else, $M$ and $E$ may be continuous variables denoting the dart's proximity to the bullseye and the size of the reward. Also note that, as this example illustrates, $M$ denotes a state brought about by a goal-directed motor command (e.g., hitting or missing a bullseye), not the motor command itself (e.g., the motor command that implements dart throwing)—an echo of ideomotor theory[28], which proposes that actions are encoded in terms of the sensory states they elicit, rather than the motor commands that generate them.

In principle, any goal-directed activity can be decomposed into means and ends, from reading a novel (where $E$ could be "discovered the protagonist's fate" and $M$ could be "finished the next chapter") to dancing a tango (where $E$ could be "impressed my partner" and $M$ could be "step forward with right foot passing left foot"). Indeed, the very definition of "goal-directed activity" stipulates the existence of a means (the activity) and an end (the goal to which the activity is directed).

Mutual information quantifies the dependence between two random variables as the degree to which observing the value of one variable reduces uncertainty (scored as entropy) about the value of the other. The mutual information between $M$ and $E$, denoted as $I(M; E)$, quantifies the degree to which observing $M$ (e.g., whether the bullseye was hit or missed) is expected to reduce uncertainty over $E$ (e.g., whether a reward will be received). $I(M; E)$ is maximized when two conditions are met: (i) before observing $M$, the value of $E$ is completely uncertain (e.g., before hitting or missing the bullseye, the probability of reward is 50%), and (ii) after observing $M$, the value of $E$ is completely certain (e.g., after hitting or missing the bullseye, the probability of reward is 100% or 0%). $I(M; E)$ is minimized when observing the value of $M$ fails to reduce any uncertainty about the outcome of $E$ (e.g., when the probability of reward is the same regardless of whether the bullseye is hit or missed).

To see how $I(M; E)$ is computed, consider two probability functions: $p_M(m)$ and $p_{E|M}(e|m)$. $p_M(m)$ specifies the subjective probability (or likelihood) of observing each possible value of $M$. If $M$ has two possible values, successful and unsuccessful, $p_M(m)$ specifies the subjective probability of performing the means successfully versus unsuccessfully. $p_{E|M}(e|m)$ specifies the subjective probability (or likelihood) of observing each possible value of $E$ conditional on each possible value of $M$. Suppose that $E$ has two possible values: attained and unattained. In this case, $p_{E|M}(e|m)$ specifies the probability of the end being attained versus unattained conditional on performing the means successfully versus unsuccessfully. Given $p_M(m)$ and $p_{E|M}(e|m)$, $I(M; E)$ can be computed:

$$I(M; E) = \mathbb{E}_{p_{E|M}(e|m)p_M(m)}\left[\log\left(\frac{p_{E,M}(e, m)}{p_E(e)p_M(m)}\right)\right] \quad (1)$$

where $p_{E,M}(e, m) = p_{E|M}(e|m)p_M(m)$ and $p_E(e) = \sum p_{E,M}(e, m)$.

According to the informational theory of flow, flow is a monotonically increasing function of $I(M; E)$. Evidence for this theory can be found in activities known to elicit flow. Consider slot machines. How do such simple devices develop such a powerful hold on so many players? Part of the answer, according to our theory, is that slot machines have very high levels of $I(M; E)$: Prior to observing the symbols on the reel, $M$, the size of the payout, $E$, is extremely uncertain, but as soon as $M$ is observed, all uncertainty is eliminated. If a slot machine's level of $I(M; E)$ was lowered, there is little doubt that flow would decline with it. Imagine a slot machine whose value of $I(M; E) = 0$. By definition, such a machine would always stop on the same

symbols, or its symbols would be unrelated to the size of its payouts. In both cases, observing $M$ would reduce zero uncertainty about $E$, and in both cases, flow would surely plummet.

Not all of our theory's predictions are intuitive. As we will see, the informational theory of flow sometimes says flow should be relatively high when factors commonly linked to flow, like skill-challenge balance (the degree to which the difficulty of one's task feels commensurate with one's ability) and controllability (the sense of being in control over one's outcomes)[1,3], are relatively low. It also assumes that flow is insensitive to variation in instrumental value, allowing for flow to persist, and even grow stronger, in the face of diminishing rewards and increasing punishments. We tested these predictions and more across five experiments.

## Results
Most experiments leveraged the "tile game," a computer-based task designed to achieve precise control over $p_M(m)$ and $p_{E|M}(e|m)$ (Fig. 1a). On each trial, a tile appears at the center of the screen for a predetermined amount of time. Participants attempt to activate the tile, making it change color, by pressing their spacebar before it disappears. Whether or not the tile is activated determines the probability of receiving a jackpot on the

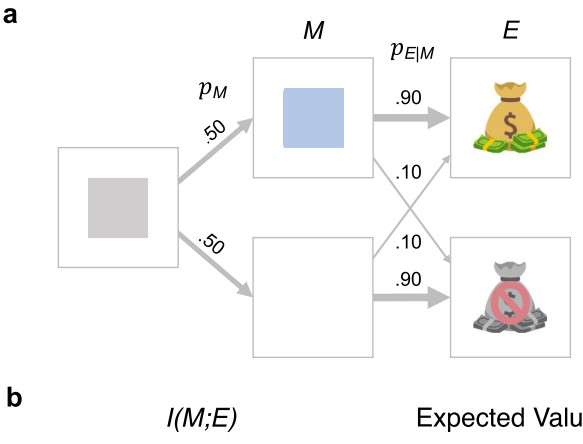

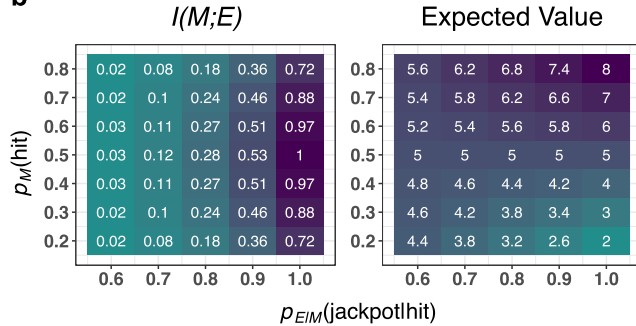

**Fig. 1 Design and features of the tile game. a** Schematic of the tile game for experiments 1–2. First, a gray tile appears on the screen, and the player must press the spacebar before it disappears. If successful, the tile changes color before transitioning to the jackpot or no jackpot state. We have illustrated the tile game with example values for $p_M(hit)$ and $p_{E|M}(jackpot|hit)$; the full set of values are given in panel b. In this example, there is a 50% chance of activating the tile, $p_M(hit) = .5$. If the tile is activated, there is a 90% chance of transitioning to the jackpot state, $p_{E|M}(jackpot|hit) = .9$, and a 10% chance of transitioning to the no jackpot state, $p_{E|M}(no\ jackpot|hit) = .1$. If the tile is not activated, there is a 10% chance of transitioning to the jackpot state, $p_{E|M}(jackpot|miss) = .1$, and a 90% chance of transitioning to the no jackpot state, $p_{E|M}(no\ jackpot|miss) = .9$. **b** Breakdown of $I(M;E)$, left, and average number of cents earned per trial (expected value), right, for different values of $p_M(hit)$ and $p_{E|M}(jackpot|hit)$.

current trial. If a player receives a jackpot, the next screen displays a pleasant image, and $0.10 are added to a bonus fund, which participants receive at the end of the study. If a jackpot is not received, the next screen displays an unpleasant image, and $0 is added to the bonus fund.

The tile game's instructions, and the design of the game itself, encourage participants to represent both $M$ and $E$ as having two possible values: $m = hit$ if the tile is activated, $m = miss$ if the tile is not activated, $e = jackpot$ if a jackpot is received, and $e = no\ jackpot$ if a jackpot is not received. For simplicity, the value of $p_{E|M}(jackpot|miss)$ is constrained to equal $1 - p_{E|M}(jackpot|hit)$. Accordingly, $I(M;E)$ is a function of two parameters: $p_M(hit)$ and $p_{E|M}(jackpot|hit)$.

Unbeknownst to participants, the tile game includes two types of trials: miss trials, where the tile disappears after 250 ms, and hit trials, where the tile disappears after 750 ms. Responding in under 250 ms is nearly impossible, and responding in under 750 ms is trivial. Thus, the percentage of hit trials corresponds to the value of $p_M(hit)$ (for analyses confirming the success of our manipulation of $p_M(hit)$ see Supplementary Information). The timing manipulation is not experienced as such. It creates the illusion of responding slightly too slow on some trials and just in time on other trials. We assume that participants tracked the value of $p_M(hit)$ either consciously or unconsciously[29,30].

To manipulate $p_{E|M}(jackpot|hit)$ and $p_{E|M}(jackpot|miss)$, we told participants the probability of attaining a jackpot conditional on activating versus not activating the tile. The true probabilities always matched these instructions. Pressing the spacebar too early produced a warning message lasting 3.5 s to disincentivize spamming the spacebar. Critically, the average amount of money obtained from the tile game is orthogonal to the value of $I(M;E)$ (Fig. 1b).

**Experiments 1 and 2.** In experiments 1 and 2, participants played two versions of the tile game (order counterbalanced), each lasting 50 trials. The games were distinguished by name and appearance: in the green game, activated tiles turned green, and in the blue game, activated tiles turned blue. For each participant and version of the game, we randomly selected $p_M(hit)$ from the set {.2, .3, .4, .5, .6, .7, .8}, and $p_{E|M}(jackpot|hit)$ from the set {.6, .7, .8, .9, 1}, with the constraint that neither parameter could be identical across both games. After each game, participants completed measures of flow, skill-challenge balance, and controllability (see Methods). Experiments 1 and 2 were identical except that experiment 2 included additional dependent measures, described below. Statistics are displayed in Figs. 2–5.

**Flow is a positive function of $I(M;E)$.** As predicted by the informational theory of flow, flow was a positive function of $I(M;E)$ in both experiments (Fig. 2a). Subsequent analyses confirmed that the effect of $I(M;E)$ on flow was positive over the full range of $I(M;E)$. Aggregating the data from both experiments, we fit a generalized additive model (GAM)—a statistical technique in which outcomes are assumed to depend on smooth, nonparametric functions of the predictors. Unlike linear regression, GAMs can discover nonlinearities that would violate the informational theory of flow. The result, however, supports our theory: The effect of $I(M;E)$ on flow was everywhere positive (Fig. 2b).

Next, we fit a GAM that modeled flow in terms of $p_M(hit)$ and $p_{E|M}(jackpot|hit)$, and generated a matrix containing the predicted value of flow for each combination of the two parameters (Fig. 2c). If flow is a positive function of $I(M;E)$, this matrix should align with the matrix representing $I(M;E)$ in terms of $p_M(hit)$ and $p_{E|M}(jackpot|hit)$ (Fig. 1b). Consistent with this, the two matrices were correlated at $r = .88$ ($p < .001$).

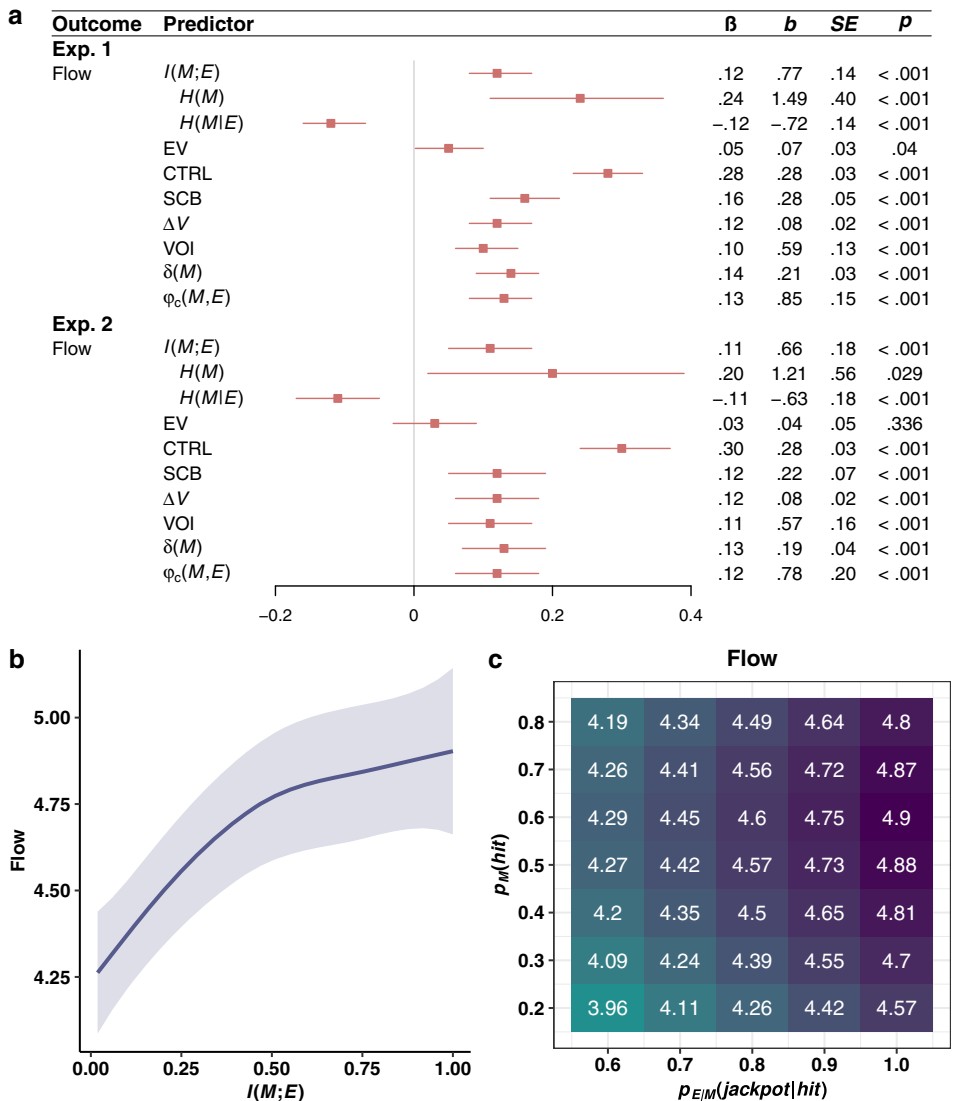

**Fig. 2 Flow results from experiments 1 and 2. a** Effect of all variables of interest on flow in experiment 1 (Exp. 1) and experiment 2 (Exp. 2). The variables of interest are $I(M;E)$, its constituent entropy terms $H(M)$ and $H(M|E)$, expected value (EV), controllability (CTRL), skill-challenge balance (SCB), marginal value ($\Delta V$), the value of information (VOI), temporal difference prediction error ($\delta(M)$), and the correlation between $M$ and $E$ ($\varphi_c(M,E)$). Statistics are derived from linear mixed models (LMMs; two-sided) performed over 720 observations across 365 participants in experiment 1, and 488 observations across 249 participants in experiment 2. Each LMM regresses flow on one variable from the predictor column (with the exception of $H(M)$ and $H(M|E)$, which are included in the same model), a nuisance regressor for game order (first game vs. second game), and random subject-level intercepts. Red squares denote standardized regression coefficients, and intersecting red lines represent 95% CIs. No corrections for multiple comparisons were applied. **b** Results of GAM showing the nonlinear, but monotonically increasing effect of $I(M;E)$ on flow. The solid line denotes expected values and the ribbons denote 95% CIs. **c** GAM-derived expected values of flow for each combination of $p_M(hit)$ and $p_{E|M}(jackpot|hit)$.

We turn now to a stricter test of our theory, one that accounts for that fact that $I(M;E)$ is a function of multiple variables, introducing the possibility that flow is not a function of $I(M;E)$ per se, but a subset of terms used to compute $I(M;E)$. Indeed, $I(M;E)$ can be expressed as:

$$I(M;E) = H(M) - H(M|E) \qquad (2)$$

where $H$ is Shannon entropy. Either entropy term on its own could fully explain a positive effect of $I(M;E)$ on flow—flow could be a positive function of $H(M)$ only, or a negative function of $H(M|E)$ only. To show that flow is a positive function of $I(M;E)$ per se, we must show that it is a positive function of $H(M)$, a negative function of $H(M|E)$, and that both effects are equivalent in magnitude. Accordingly, we simultaneously regressed flow on $H(M)$ and $H(M|E)$. Equation 2 is one of several formulations of

$I(M;E)$, but two features make it uniquely suited to our analytic approach: it describes a linear function, and the correlation between its variables, $H(M)$ and $H(M|E)$, is small enough to adhere to the multicollinearity assumption (experiment 1: $r = .179$; experiment 2: $r = .166$).

Our entropy-based analyses suggest that flow is a positive function of $I(M;E)$ per se rather than individual terms used to compute it (Fig. 2a). $H(M)$ had positive effects, $H(M|E)$ had negative effects, we could not reject the hypothesis that these effects are equivalent in magnitude (experiment 1: $\chi^2(1) = 3.59$, $p = .058$; experiment 2: $\chi^2(1) = 1.13$, $p = .287$), and the Bayesian information criterion (BIC) favored models that treated flow as a function of $I(M;E)$ (experiment 1: BIC = 2731; experiment 2: BIC = 1868) over models that treated flow as a function of $H(M)$ and $H(M|E)$ (experiment 1: BIC = 2734; experiment 2: BIC = 1873).

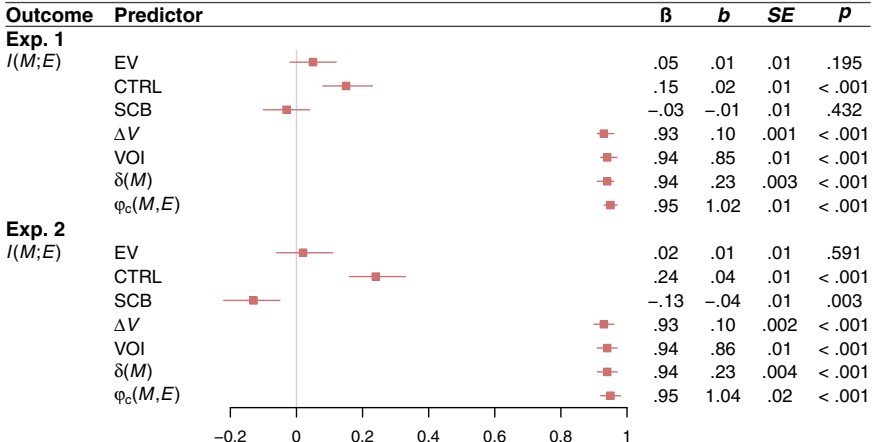

**Fig. 3 Relationships between $I(M;E)$ and variables of interest in experiments 1 and 2.** Effects of all variables of interest on $I(M;E)$ in experiment 1 (Exp. 1) and experiment 2 (Exp. 2). The variables of interest are expected value (EV), controllability (CTRL), skill-challenge balance (SCB), marginal value ($\Delta V$), the value of information (VOI), temporal difference prediction error ($\delta(M)$), and the correlation between $M$ and $E$ ($\varphi_c(M,E)$). Statistics are derived from LMMs (two-sided) performed over 720 observations across 365 participants in experiment 1, and 488 observations across 249 participants in experiment 2. Each LMM regresses $I(M;E)$ on one variable from the predictor column, a nuisance regressor for game order (first game vs. second game), and random subject-level intercepts. Red squares denote standardized regression coefficients, and intersecting red lines represent 95% CIs. No corrections for multiple comparisons were applied.

The effects of $I(M;E)$ on flow cannot be explained in terms of expected value or skill-challenge balance. $I(M;E)$ was uncorrelated with skill-challenge balance in experiment 1, negatively correlated with skill-challenge balance in experiment 2, and uncorrelated with expected value in both experiments (Fig. 3). Adjusting for expected value or skill-challenge balance never eliminated the effect of $I(M;E)$ on flow (see Supplementary Information).

**$I(M;E)$, enjoyment, and attention.** Flow often coincides with enjoyment and improved attentional performance[1,3,6,31–33], so we measured these outcomes in experiment 2. Showing that $I(M;E)$ predicts enjoyment or attention would bolster our claim that flow increases with greater $I(M;E)$ while demonstrating that $I(M;E)$ can deliver benefits beyond the subjective experience of immersion.

**Enjoyment.** We collected two measures of enjoyment: a continuous self-report measure (administered after each game), and a binary choice measure that asked participants which game they would prefer to play again (administered at the end of the experiment; see Methods). The effect of $I(M;E)$ on the continuous measure was positive but not significant (Fig. 4). However, $I(M;E)$ had a significant effect on the binary choice measure such that the greater the value of $I(M;E)$ in one game versus the other, the likelier that game was to be chosen (Fig. 4). This finding supports the idea that $I(M;E)$ predicts enjoyment, providing converging evidence for the informational theory of flow.

**Attention.** In the tile game, greater attention should make responses to the tiles faster[34,35] and less variable[36–39]. Thus, in experiment 2, we recorded response times. We operationalized attention as the average time between the onset of the gray tile and the pressing of the spacebar (RT), and the intra-individual standard deviation of response times (RTSD). RTSD reflects attentional lapses and distractibility, and has been linked to attentional impairments[36–39]. For example, individuals with attention deficit hyperactivity disorder (ADHD) exhibit significantly greater RTSD[40]. If attention increases with greater $I(M;E)$, then RT and RTSD should decrease with greater $I(M;E)$.

Increasing $I(M;E)$ improved attentional performance as revealed by significant, negative effects on RT and RTSD (Fig. 5a). Moreover, GAMs revealed that these effects were monotonically decreasing (Fig. 5b, c), providing further support for the idea that the effect of $I(M;E)$ on flow is monotonically increasing. These findings provide converging evidence for our theory while demonstrating its utility for optimizing attentional performance.

As expected, both measures of attention varied not only with $I(M;E)$, but also with $p_M(hit)$ (Supplementary Information). Increasing $p_M(hit)$ increases RT and RTSD by reducing the average speed with which participants must respond. Critically, the linear effect of $p_M(hit)$ cannot account for the linear effect of $I(M;E)$ because $I(M;E)$ is quadratic with respect to $p_M(hit)$ (Fig. 1a). Nonetheless, we included $p_M(hit)$ as a covariate in all analyses of RT and RTSD. Removing this covariate had no meaningful effect on our results (Supplementary Information).

**Confounds.** Experiments 1 and 2 include several confounds. Both $I(M;E)$ and flow were positively associated with the following variables: marginal value, the value of information, temporal difference prediction error, the correlation between $M$ and $E$, and controllability (Figs. 2a and 3). Below, we define each variable before ruling them out in experiment 3.

**Marginal value.** Marginal value, denoted as $\Delta V$, is the average reward obtained for activating versus not activating the tile:

$$\Delta V = \sum_e U(e) * p_{E|M}(e|hit) - \sum_e U(e) * p_{E|M}(e|miss) \quad (3)$$

where $U(e)$ is the financial outcome for each value of $E$. Intuitively, $\Delta V$ quantifies how much better it is to perform the means successfully versus unsuccessfully.

**Value of information.** The value of information, or VOI, quantifies the degree to which information can be used to increase expected future rewards. For instance, on each trial of the tile game, participants obtain information about their probability of obtaining jackpots, and can use this information to decide if it is more lucrative to continue playing or to quit the experiment early to pursue different activities. A recent theoretical analysis proposed that VOI may promote flow[41].

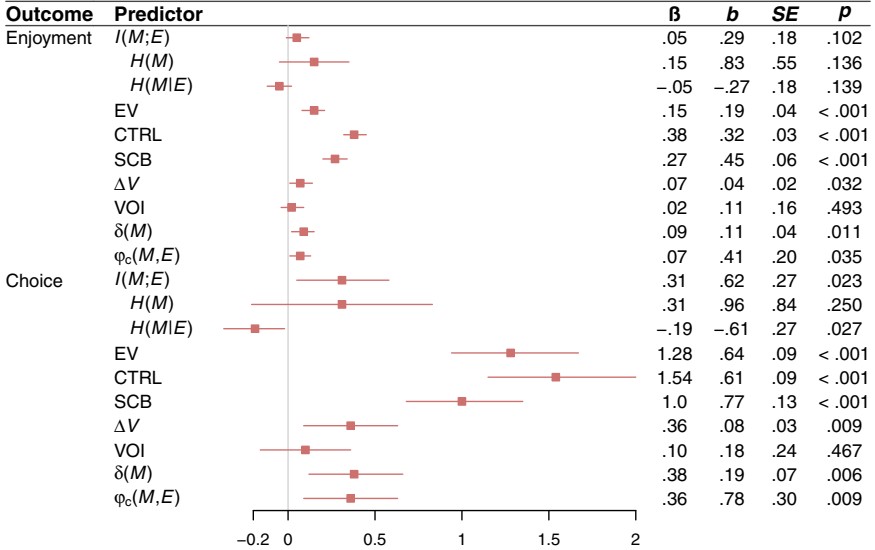

| Outcome | Predictor | | ß | b | SE | p |
|---|---|---|---|---|---|---|
| Enjoyment | $I(M;E)$ | | .05 | .29 | .18 | .102 |
| | $H(M)$ | | .15 | .83 | .55 | .136 |
| | $H(M|E)$ | | −.05 | −.27 | .18 | .139 |
| | EV | | .15 | .19 | .04 | < .001 |
| | CTRL | | .38 | .32 | .03 | < .001 |
| | SCB | | .27 | .45 | .06 | < .001 |
| | $\Delta V$ | | .07 | .04 | .02 | .032 |
| | VOI | | .02 | .11 | .16 | .493 |
| | $\delta(M)$ | | .09 | .11 | .04 | .011 |
| | $\varphi_c(M,E)$ | | .07 | .41 | .20 | .035 |
| Choice | $I(M;E)$ | | .31 | .62 | .27 | .023 |
| | $H(M)$ | | .31 | .96 | .84 | .250 |
| | $H(M|E)$ | | −.19 | −.61 | .27 | .027 |
| | EV | | 1.28 | .64 | .09 | < .001 |
| | CTRL | | 1.54 | .61 | .09 | < .001 |
| | SCB | | 1.0 | .77 | .13 | < .001 |
| | $\Delta V$ | | .36 | .08 | .03 | .009 |
| | VOI | | .10 | .18 | .24 | .467 |
| | $\delta(M)$ | | .38 | .19 | .07 | .006 |
| | $\varphi_c(M,E)$ | | .36 | .78 | .30 | .009 |

**Fig. 4 Enjoyment results from experiment 2.** Effects of all variables of interest on the continuous measure of enjoyment and the binary choice measure of which game participants would rather play again in experiment 2. The variables of interest are $I(M;E)$, its constituent entropy terms $H(M)$ and $H(M|E)$, expected value (EV), controllability (CTRL), skill-challenge balance (SCB), marginal value ($\Delta V$), the value of information (VOI), temporal difference prediction error ($\delta(M)$), and the correlation between $M$ and $E$ ($\varphi_c(M,E)$). Effects on choice are coded such that positive scores correspond to a tendency to choose the game with the highest value of the relevant variable. Statistics are derived from LMMs (two-sided) performed over 488 observations across 249 participants. Each LMM regresses one variable from the outcome column on one variable from the predictor column (with the exception of $H(M)$ and $H(M|E)$, which are included in the same model), a nuisance regressor for game order (first game vs. second game), and random subject-level intercepts. Red squares denote standardized regression coefficients, and intersecting red lines represent 95% CIs. No corrections for multiple comparisons were applied.

Accordingly, we computed VOI for each combination of $p_M(hit)$ and $p_{E|M}(jackpot|hit)$ (see Methods).

**Temporal difference prediction error.** According to computational models of reinforcement learning, reward-predicting stimuli elicit learning signals called temporal difference prediction errors, which humans use to estimate long-run future rewards[42–44]. On each trial of the tile game, the value of $M$ acts as a reward-predicting stimulus by indicating the probability of a jackpot. Therefore, we assume that on each observation of $M$, participants encoded a temporal difference prediction error, denoted as $\delta(M)$. This variable quantifies the degree to which observing $M$ increases or decreases the amount of money a participant expects to earn on a given trial. We computed $\delta(M)$ for each combination of $p_M(hit)$ and $p_{E|M}(jackpot|hit)$ (see Methods).

**Correlation.** The correlation between $M$ and $E$ is quantified in terms of Cramér's phi, and is denoted as $\varphi_c(M,E)$.

**Controllability.** We operationalized controllability as the degree to which participants reported feeling in control of their outcomes during the tile game (see Methods). Interestingly, previous work has equated controllability with an information theoretic quantity similar to $I(M;E)$, implying that controllability and $I(M;E)$ may be impossible to disentangle[45,46]. Challenging this idea, we successfully separated controllability from $I(M;E)$ in experiment 3.

**Experiment 3.** We address each of the above confounds in experiment 3. All participants played two versions of the tile game in counterbalanced order: the mixture game and either the punish game or the neutral game (Fig. 6a–c). After each game,

we measured flow, enjoyment (using the continuous measure from experiment 2), attention (in terms of RT and RTSD), controllability, and skill-challenge balance.

In all three games, $p_M(hit) = .5$, and $p_{E|M}(jackpot|hit) = 1$. The main difference between each game is the outcome of a miss. In the neutral game, misses always yield $0; in the punish game, misses always yield a $0.02 loss; in the mixture game, misses always result in a fifty-fifty chance of $0 or a $0.02 loss. $I(M;E)$ is greatest in the mixture game and identical across the other games (Fig. 6d). This is not true of expected value, skill-challenge balance, or any of the confounds described above (Fig. 6d). Thus, the informational theory of flow uniquely predicts that flow should be highest in the mixture game and identical in the other games. This is a surprising prediction. The basic principle that organisms aim to avoid punishment[47–49] suggests that flow should be greatest in the neutral game, where punishment is least frequent. Conversely, the principle that the attentional system prioritizes punishment-related stimuli[50] suggests that flow should be greatest in the punish game, where punishment is most frequent. We know of no theory besides ours that expects flow to be greatest in the mixture game, where punishment is neither least frequent nor most frequent.

**Flow is a positive function of $I(M;E)$.** We regressed flow on game (mixture vs. punish vs. neutral). As predicted, the effect of game was significant ($\chi^2(2) = 16.68$, $p < .001$; Fig. 7) such that flow was greatest in the mixture game (mixture vs. punish: $b = .29$, $SE = .1$, $p = .005$; mixture vs. neutral: $b = .33$, $SE = .1$, $p = .002$), and equivalent across the punish and neutral games ($b = .04$, $SE = .14$, $p = .801$). These findings support the informational theory of flow, and cannot be explained in terms of $\Delta V$, VOI, $\delta(M)$, $\varphi_c(M,E)$, controllability, skill-challenge balance, or expected value.

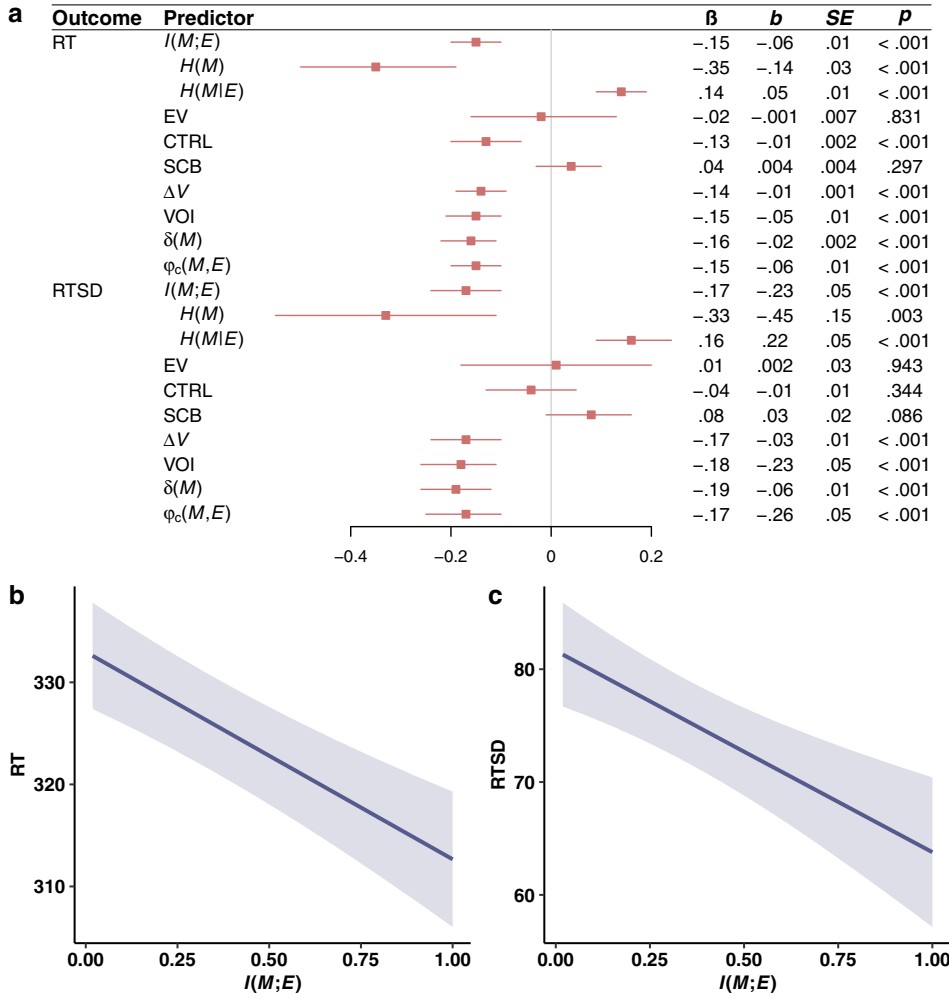

**Fig. 5 Attention results from experiment 2. a** Effects of all variables of interest on attentional performance in experiment 2, operationalized as response time (RT) and the standard deviation of response times (RTSD). Lower values of RT and RTSD correspond to greater attention. RT and RTSD were log transformed prior to analysis to correct for skewness. The variables of interest are $I(M;E)$, its constituent entropy terms $H(M)$ and $H(M|E)$, expected value (EV), controllability (CTRL), skill-challenge balance (SCB), marginal value ($\Delta V$), the value of information (VOI), temporal difference prediction error ($\delta(M)$), and the correlation between $M$ and $E$ ($\varphi_c(M,E)$). Statistics are derived from LMMs (two-sided) performed over 488 observations across 249 participants. Each LMM regresses one variable from the outcome column on one variable from the predictor column (with the exception of $H(M)$ and $H(M|E)$, which are included in the same model), a nuisance regressor for game order (first game vs. second game), a nuisance regressor for $p_M(hit)$ to control for task demands on response time, and random subject-level intercepts. Red squares denote standardized regression coefficients, and intersecting red lines represent 95% CIs. No corrections for multiple comparisons were applied. **b** Results of GAM showing the monotonically decreasing effect of $I(M;E)$ on RT in units of milliseconds. The solid line denotes expected values and the ribbons denote 95% CIs. **c** Results of GAM showing the monotonically decreasing effect of $I(M;E)$ on RTSD in units of milliseconds. The solid line denotes expected values and the ribbons denote 95% CIs.

**$I(M;E)$ and attention**. Game had no effect on attention (RT: $\chi^2(2) = 3.37$, $p = .185$; RTSD: $\chi^2(2) = 2.97$, $p = .227$), likely due to a floor effect. The smallest value of $I(M;E)$ in experiment 3 is equal to the largest value of $I(M;E)$ from experiments 1 and 2, and increasing $I(M;E)$ beyond this point may have diminishing returns on RT and RTSD due to physical limitations; eventually, people cannot respond any faster or less variably[51]. Inspection of the RT distribution confirmed that participants were close to floor: 33% of mean RTs were 300 ms or less and the median was 320 ms. For reference, when analyzing RT data, researchers often exclude participants with mean RTs of 300 ms or less because such participants are considered extreme outliers[52–54]. For the lowest values of $I(M;E)$ in experiment 2 (those in the bottom quartile), RTs were significantly slower: only 19% of mean RTs were 300 ms or less ($\chi^2(1) = 7.69$, $p = .006$) and the median was 336 ms (Wilcoxon rank-sum test, $p = .003$), providing room for faster responding not present in experiment 3.

**$I(M;E)$ and enjoyment**. Converging support for the informational theory of flow comes from analyses of enjoyment. The effect of game was significant ($\chi^2(2) = 12.5$, $p = .002$; Fig. 7), such that participants enjoyed the mixture game significantly more than the punish game ($b = .34$, $SE = .1$, $p < .001$) and non-significantly more than the neutral game ($b = .13$, $SE = .1$, $p = .195$). We found no significant difference in enjoyment across the punish and neutral games ($b = .21$, $SE = .14$, $p = .134$). It is noteworthy that the neutral game, which does not involve punishment, was not enjoyed more than the mixture game, which does involve punishment. Apparently, the effect of $I(M;E)$ on enjoyment is powerful enough to overcome aversion to negative outcomes.

**Experiment 4**. If instead of playing the tile game participants merely observed it, would the mutual information between the

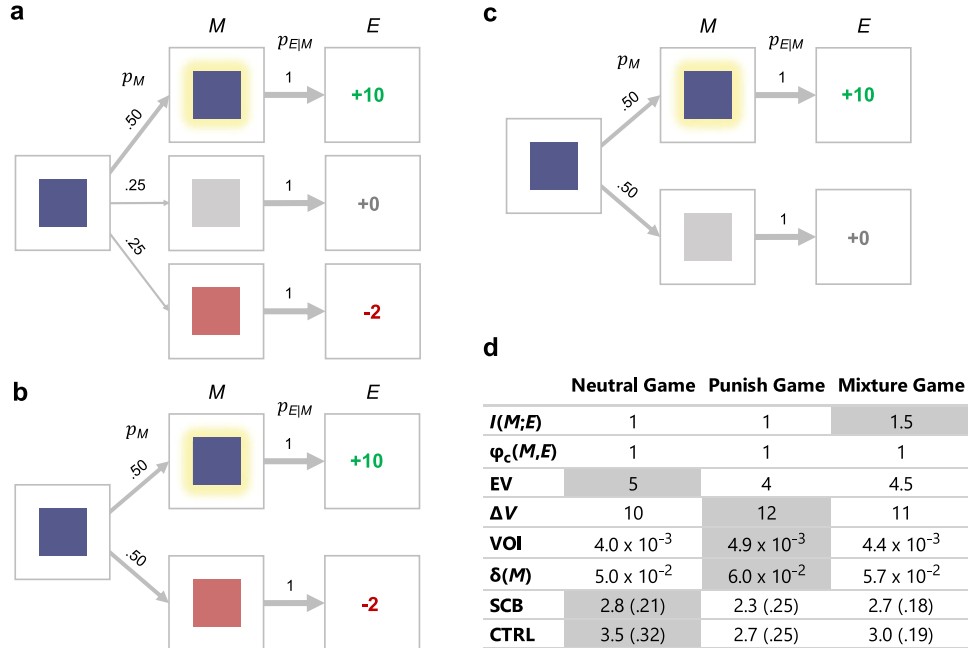

| | Neutral Game | Punish Game | Mixture Game |
|---|---|---|---|
| $I(M;E)$ | 1 | 1 | 1.5 |
| $\varphi_c(M,E)$ | 1 | 1 | 1 |
| EV | 5 | 4 | 4.5 |
| $\Delta V$ | 10 | 12 | 11 |
| VOI | $4.0 \times 10^{-3}$ | $4.9 \times 10^{-3}$ | $4.4 \times 10^{-3}$ |
| $\delta(M)$ | $5.0 \times 10^{-2}$ | $6.0 \times 10^{-2}$ | $5.7 \times 10^{-2}$ |
| SCB | 2.8 (.21) | 2.3 (.25) | 2.7 (.18) |
| CTRL | 3.5 (.32) | 2.7 (.25) | 3.0 (.19) |

**Fig. 6 Design of experiment 3.** In experiment 3, each version of the tile game shared the following features: (i) the tile glowed when activated (instead of changing colors, as in experiments 1 and 2), (ii) $p_M(hit)$ was always set to .5, and (iii) $p_{E|M}(jackpot|hit)$ was always set to 1. **a** Mixture game. Missing the tile results in a 50% chance of the tile turning red, indicating a $0.02 loss (i.e., a deduction of $0.02 from the player's bonus fund), and a 50% chance of the tile turning gray (indicating no jackpot); $M$ has three possible values (*hit, gray,* and *red*) and $E$ has three possible values (*jackpot, no jackpot,* and *penalty*). **b** Punish game. Missing the tile results in a 100% chance of the tile turning red (indicating a $0.02 less); $M$ has two possible values (*hit* and *red*) and $E$ has two possible values (*jackpot* and *penalty*). **c** Neutral game. Missing the tile results in a 100% chance of the tile turning gray, indicating no jackpot; $M$ has two possible values (*hit* and *gray*) and $E$ has two possible values (*jackpot* and *no jackpot*). **d** Key statistics associated with each version of the game. The values of $I(M;E)$, expected value (EV), marginal value ($\Delta V$), and the correlation between $M$ and $E$ ($\varphi_c(M,E)$) were computed analytically. The values for temporal difference prediction error ($\delta(M)$) and value of information (VOI) correspond to the average output of 1,000 simulations. The values of skill-challenge balance (SCB) and controllability (CTRL) are the empirical means and standard errors obtained from self-report measures of the corresponding constructs. The largest value of each variable is highlighted in gray.

tile state (hit vs. miss) and jackpot state (jackpot vs. no jackpot) still predict flow[55,56]? Our theory says it would not. The $M$ in $I(M;E)$ denotes a state that someone brings about to achieve their goal, and someone who merely observes the tile game would not bring about the tile state. Thus, for observers, the tile state would not correspond to $M$, and the mutual information between tile state and jackpot state would not correspond to $I(M;E)$. Since our theory says that flow is a function of $I(M;E)$ specifically, not mutual information generally, it predicts that when the tile game is merely observed, the mutual information between the tile state and jackpot state does not predict flow.

We tested this prediction by assigning each participant to a play condition, where they played the tile game from experiments 1 and 2, or an observe condition, where they merely observed the tile game (see Methods). Participants played or observed two games following the procedures from experiments 1 and 2. Jackpots were worth 1 cent. After each game, we measured flow and enjoyment. RT-based measures of attention were collected only in the play condition, as the observe condition prohibits responding. Analyses of enjoyment and attention unanimously supported the informational theory of flow (Fig. 8).

**Flow is a positive function of $I(M;E)$ specifically, not mutual information generally.** Due to a coding error, the randomly selected values of $p_M(hit)$ and $p_{E|M}(jackpot|hit)$ were not recorded in the play condition. Accordingly, we computed the empirical value of $I(M;E)$ using the values of $p_M(hit)$ and $p_{E|M}(jackpot|hit)$ actually produced by participant. (In experiments 1 and 2, the empirical value of $I(M;E)$ was almost perfectly correlated with

the true value at $r > .98$, and all analyses produced equivalent results regardless of which value we used).

We regressed flow on condition (play vs. observe), mutual information, and their interaction term. As predicted, we found a significant interaction ($b = .43$, $SE = .17$, $p = .013$) such that flow was a positive function of mutual information among players but not observers (Fig. 8), suggesting that flow is a positive function of $I(M;E)$ specifically, not mutual information generally. Further support for the informational theory of flow comes from our entropy-based analyses (Fig. 8): In the play condition, flow was a positive effect of $H(M)$ and a negative effect of $H(M|E)$, and we found no evidence that these effects differ in magnitude ($\chi^2(1) = .74$, $p = .391$).

**Experiment 5**. In experiment 5, we generalize the informational theory of flow to new tasks. Instead of creating tasks, we subjected our theory to the critical test of making correct predictions about existing activities developed without any intention of supporting our theory. The activities we chose are two of the world's oldest games: Rock, Paper, Scissors, known as *shoushiling* in the time of the Chinese Han dynasty[57], and Odds vs. Evens, known as *ludere par impar* in ancient Rome[58] (Fig. 9A, B). In Rock, Paper, Scissors, $I(M;E) = 1.58$, and in Odds vs. Evens, $I(M;E) = 1$ (see Methods), so our theory predicts that Rock, Paper, Scissors elicits more flow. This finding would further rule out four alternative constructs: $\Delta V$, VOI, or $\delta(M)$, which are higher in Odds vs. Evens, and $\varphi_c(M, E)$, which is identical across both games (Fig. 9d). The games in experiment 5 bear little resemblance to the tile game. For instance, the tile game is presented as a game of

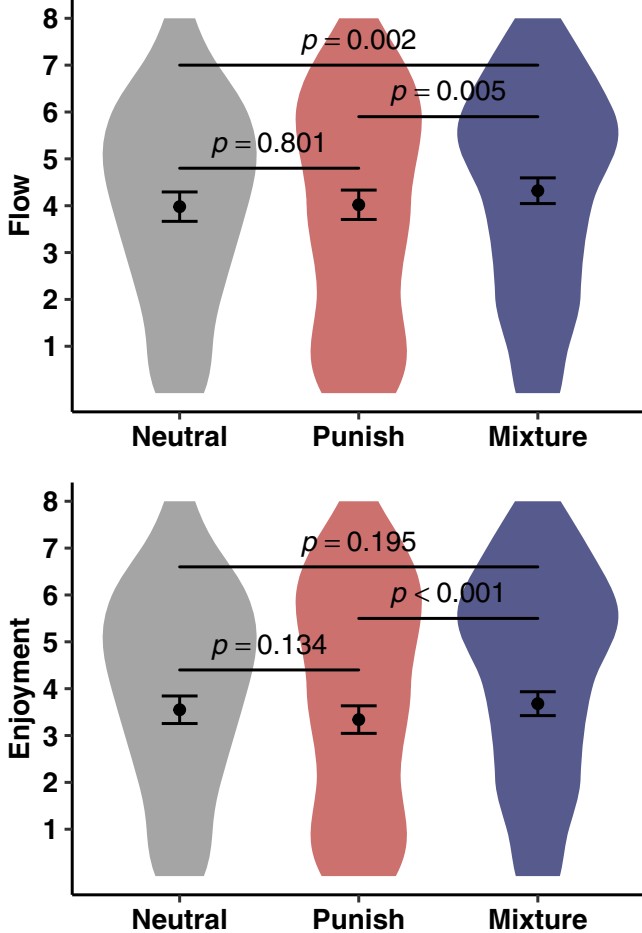

**Fig. 7 Results of experiment 3.** Effect of game on flow and enjoyment in experiment 3. Points denote means and error bars represent 95% CIs. Statistics are derived from LMMs (two-sided) performed over 479 observations across 244 participants. No corrections for multiple comparisons were applied.

skill, whereas the hand games are games of chance — players cannot benefit financially from trying harder or paying more attention. This change eliminates features that, according to prior research, promote feelings of agency and control[59,60], which some consider important for flow[3].

Participants played 50 rounds of both games against a computer that chose hands at random. Participants earned $0.03 for every win and lost $0.01 for every loss. Draws, which are possible only in Rock, Paper, Scissors, were worth $0. After each game, we measured flow and enjoyment.

**The informational theory of flow generalizes.** Rock, Paper, Scissors elicited significantly more flow ($b = .59$, $SE = .08$, $p < .001$) and enjoyment ($b = .59$, $SE = .07$, $p < .001$) than Odds vs. Evens (Fig. 9c). These findings confirm that the informational theory of flow generalizes beyond the task originally designed to test it, and further rules out $\Delta V$, VOI, $\delta(M)$, and $\varphi_c(M, E)$ alternative constructs.

## Discussion

Flow is considered a key contributor to health, productivity, and well-being[1–3,10,11,31], yet a rigorous description of its computational substrates remains elusive. To understand the nature of flow, and to help people regulate flow in their daily lives, it is necessary to ground the concept of flow in formal theoretical

structures. We developed such a structure — the informational theory of flow — and obtained empirical support for it across five experiments by showing that flow, along with enjoyment and attention, increases as a function of $I(M; E)$.

$I(M; E)$, it seems, is an important contributor to flow—but why? What do humans have to gain by becoming immersed when $I(M; E)$ is high? We speculate that the link between $I(M; E)$ and flow facilitates the fundamental task of learning associations between actions and desired outcomes. This task is complicated by the fact that every desired outcome (e.g., sating hunger) is associated with a relatively small number of actions (e.g., eating)—most action-outcome pairs are unrelated (e.g., sating hunger and knitting). We must restrict our learning efforts to the subset of valid action-outcome pairs. Otherwise, we risk wasting resources trying to learn associations that do not exist. One way to meet this challenge is to become immersed when $I(M; E)$ is high, and grow bored when $I(M; E)$ is low. Indeed, the greater the value of $I(M; E)$, the more information can be gained by learning the relationship between $M$ and $E$. Accordingly, the positive effect of $I(M; E)$ on flow may serve the function of steering us toward learning opportunities and away from epistemic dead ends.

Another open question is how human brains might compute $I(M; E)$. When considering the complexities of many real-world activities, such as extended action sequences and hierarchical structure, it becomes clear that computing $I(M; E)$ exactly is intractable. Accordingly, brains likely implement algorithms that quickly and efficiently approximate $I(M; E)$. Examples of such algorithms are emerging in the AI literature[21,23] and may serve as inspiration for biologically plausible implementations of the informational theory of flow.

Two caveats deserve spotlighting. First, the present work does not suggest that $I(M; E)$ is the sole contributor to flow, nor does it suggest that $I(M; E)$ contributes to flow across all contexts. The informational theory of flow may yet be expanded by discoveries of additional inputs to the flow-generating process, and contracted by discoveries of contexts in which $I(M; E)$ fails to predict flow. A second caveat is that the quantity at the heart of our theory — $I(M; E)$ — is a function of variables whose properties are subjective. What $M$ and $E$ denote in a given task depends on how the person performing the task construes their means and end. On the one hand, the perceiver-dependence of $M$ and $E$ allows our theory to explain individual and situational differences in how much flow a particular activity elicits. On the other hand, it makes our theory difficult to apply to tasks with many possible means-end representations, a challenge we overcame by using tasks with clear means and ends. Expanding our theory to more ambiguous tasks hinges on the progress of ongoing research exploring how humans represent task structure[27,61–64]. With better theories of how humans carve activities into means and ends, the informational theory of flow will become easier for researchers to falsify and for practitioners to apply.

In addition to raising challenging new questions, the present work supports the recent movement to give computational tools a more prominent role in social psychology[65]. As others have argued, grounding social psychological phenomena in formal theoretical structures can help us deliver more robust and replicable solutions to societal challenges. This idea has started to take hold, leading social psychologists to import formalisms from a variety of frameworks, such as reinforcement learning, probability theory, and utility theory[65]. We continue this trend by importing the formalism of mutual information — a concept from information theory, which remains underutilized in social psychology. In this way, the informational theory of flow offers insights into task immersion and engagement, and expands the conceptual toolkit for social psychological theorizing and model building.

| Outcome | Predictor | | ß | b | SE | p |
|---|---|---|---|---|---|---|
| **Play** | | | | | | |
| Flow | $I(M;E)$ | | .06 | .44 | .12 | < .001 |
| | $H(M)$ | | .11 | .77 | .37 | .038 |
| | $H(M|E)$ | | −.06 | −.43 | .12 | < .001 |
| Enjoyment | $I(M;E)$ | | .05 | .31 | .13 | .016 |
| | $H(M)$ | | .18 | 1.17 | .39 | .003 |
| | $H(M|E)$ | | −.04 | −.27 | .13 | .037 |
| RT | $I(M;E)$ | | −.12 | −.06 | .01 | < .001 |
| | $H(M)$ | | −.19 | −.10 | .03 | .004 |
| | $H(M|E)$ | | .12 | .06 | .01 | .005 |
| RTSD | $I(M;E)$ | | −.10 | −.13 | .04 | < .001 |
| | $H(M)$ | | −.03 | −.05 | .11 | .644 |
| | $H(M|E)$ | | .10 | .13 | .04 | < .001 |
| **Observe** | | | | | | |
| Flow | $I(M;E)$ | | .002 | .01 | .12 | .915 |
| | $H(M)$ | | .05 | .37 | .36 | .295 |
| | $H(M|E)$ | | .001 | .02 | .12 | .963 |
| Enjoyment | $I(M;E)$ | | .004 | .02 | .13 | .857 |
| | $H(M)$ | | .08 | .55 | .38 | .144 |
| | $H(M|E)$ | | .001 | .004 | .13 | .971 |

−0.2    0    0.2

**Fig. 8 Results of experiment 4.** Effects of $I(M;E)$ and its component entropy terms, $H(M)$ and $H(M|E)$, on flow, enjoyment, RT, and RTSD in the both conditions (play and observe) of experiment 4. Statistics are derived from LMMs (two-sided) performed over 1882 observations across 941 participants. Each LMM regresses one variable from the outcome column on one variable from the predictor column (with the exception of $H(M)$ and $H(M|E)$, which are included in the same model), a nuisance regressor for game order (first game vs. second game), and random subject-level intercepts. LMMs predicting RT and RTSD also include a nuisance regressor for $p_M(hit)$ to control for task demands on response time. Red squares denote standardized regression coefficients, and intersecting red lines represent 95% CIs. No corrections for multiple comparisons were applied.

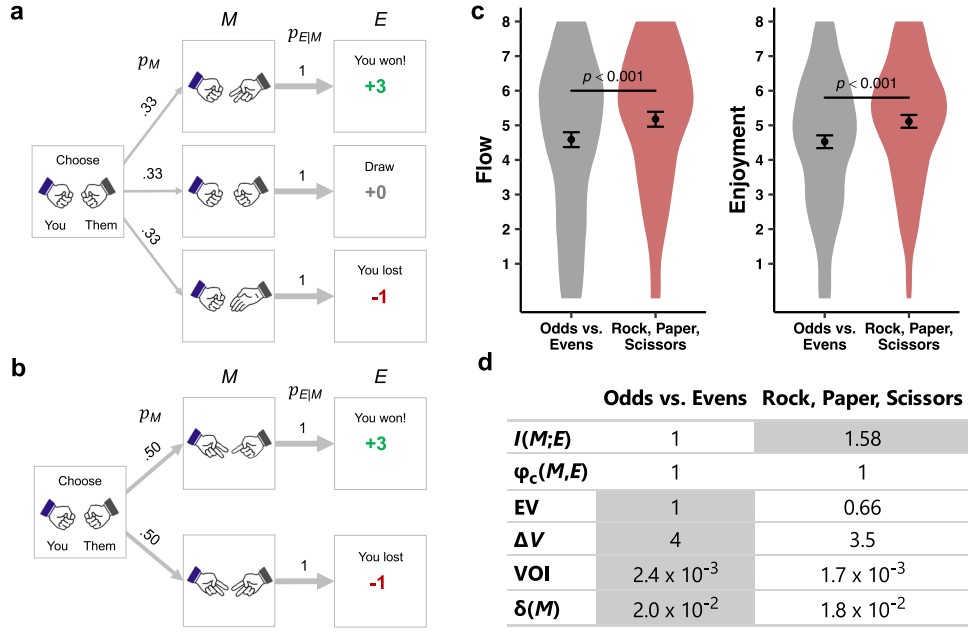

**Fig. 9 Design and results of experiment 5. a** Schematic of Rock, Paper, Scissors. On each trial, participants choose one of three symbols (rock, paper, or scissors), then transition to one of three states depending on their (computer) opponent's symbol (rock, paper, or scissors) with a 33% chance of each state. Depending on the combination of symbols, the end state is a win (worth 3 cents), draw (worth 0 cents), or loss (worth -1 cent). We have illustrated the game with the example choice of "rock" for the participant. **b** Schematic of Odds vs. Evens. Participants first chose to represent "odds" or "evens" for the duration of the game. On each trial, participants select a number (one or two), then transition to one of two states depending on the number selected by their (computer) opponent (one or two), with a 50% chance of each state. If the sum of the numbers is odd, participants advance to an end state of "win" (worth 3 cents) if they represent "odds"—otherwise they advance to an end state of "lose" (worth -1 cents). If the sum of the numbers if even, participants advance to an end state of "win" if they represent "evens"—otherwise they advance to an end state of "lose." We have illustrated the game for the example choice of two for a participant who represents "odds." **c** Effects of game on flow and enjoyment. Points denote means, and error bars represent 95% CIs. Statistics are derived from LMMs (two-sided) performed over 794 observations across 397 participants. No corrections for multiple comparisons were applied. **d** Key statistics associated with each game. The values of $I(M;E)$, expected value (EV), marginal value ($\Delta V$), and the correlation between $M$ and $E$ ($\varphi_c(M,E)$) were computed analytically. The values for temporal difference prediction error ($\delta(M)$) and value of information (VOI) correspond to the average output of 1,000 simulations. The largest value of each variable is highlighted in gray.

## Methods

This research was approved by the Human Subjects Committee of Yale University, New Haven, CT, USA. All participants provided informed consent and were compensated for their time. We preregistered experiments 2 (https://aspredicted.org/cx6up.pdf), 3 (https://aspredicted.org/m3ez6.pdf), 4 (https://aspredicted.org/ef7g5.pdf) and 5 (https://aspredicted.org/y4wx9.pdf). Based on pilot data, we expected $I(M; E)$ to have a small-to-medium sized effect on flow ($r = .2$), requiring a sample of 194 participants to achieve 80% power for two-sided tests. Thus, we aimed for a final sample size of at least 200 in each experiment. We recruited participants using Prolific and recorded data using jsPsych (version 6.1.0). Unless otherwise specified, all analyses were conducted using linear mixed models (LMMs) with subject-level random intercepts fit using the lme4 package in RStudio version 1.3 using R version 3.6, and game order (first game vs. second game) was included as a nuisance regressor.

**Participants**. We recruited 400 participants in experiment 1, 300 participants in experiments 2 and 3, 1000 in experiment 4, and 400 in experiment 5. Our final samples after exclusions included N = 365 in experiment 1 (62% female; mean age = 35), N = 249 in experiment 2 (68% female; mean age = 26), N = 236 in experiment 3 (52% female; mean age = 32), N = 941 in experiment 4 (59% female; mean age = 40), and N = 397 in experiment 5 (60% female; mean age = 37).

**Data processing and multiverse analysis**. Participants were excluded if any of the following conditions were met: (i) they failed to answer every self-report question, (ii) during at least one tile game, they activated the tile at least five times more or less than they should have given the value of $p_M(hit)$, or (iii) during at least one tile game, they pressed their spacebar preemptively (i.e. before the tile appeared) at least 10 times. In experiment 4, we needed a sufficient number of hit trials to estimate $I(M; E)$, so we excluded participants who did not have at least five (N = 11). When calculating mean RT and RTSD, we excluded extremely fast responses (RT < 100 ms) and responses that came immediately after a preemptive response (preemptive responses were followed by an attention-grabbing warning message — "TOO FAST!" — which may influence responding on the following trial). All exclusion criteria were chosen prior to data analysis, but in some cases, they deviated from our preregistrations. Accordingly, we performed a multiverse analysis in which we applied all possible permutations of exclusion criteria, data transformation (i.e., log transforming versus not), and model specification (e.g., including versus not including covariates) (see Supplementary Information). Significant effects in the main text remained significant across the vast majority of the multiverse; marginal effects in the main text were, predictably, less robust. In general, the most reliable effects were those of $I(M; E)$ on flow and attention. Effects of $I(M; E)$ on enjoyment were more sensitive to data processing decisions.

**Flow**. We measured flow immediately after each tile game was completed. In experiments 1–3, we asked how immersive, engaging, engrossing, and addictive the game was. These items were adopted from existing measures of flow[66]. Participants responded on a 9-point scale from 1 = Not at all to 9 = Extremely. A principal components analyses with varimax rotation suggested that the "addictive" item did not load on the same factor as "immersive," "engaging," and "engrossing" (Supplementary Information, Supplementary Table 1). However, all significant results remained significant, and all non-significant results remained non-significant, when analyses were run with a three-item measure comprised of the "immersive," "engaging," and "engrossing" items. In experiments 4 and 5, we replaced the "addictive" item with an item that asked how absorbing the game was. Cronbach's α for the flow measure was at least .92 in all studies.

**Skill-challenge balance**. Participants rated skill-challenge balance by answering the question, "Was the [green game / blue game] too easy, too hard, or just the right level of difficulty?" on a 9-point scale from 0 = "Way too easy" to 8 = "Way too hard". Scores were computed by subtracting the absolute value of the distance from "Just right" from four.

**Controllability**. Participants rated controllability by answering the question, "While playing the [green game / blue game], how much control did you feel you had over the outcome of each trial?" on a 9-point scale from 0 = "Zero control" to 8 = "Complete control".

**Enjoyment**. The continuous measure of enjoyment consisted of five items. Participants indicated how enjoyable, fun, and entertaining the game was on a 9-point scale from 1 = Not at all to 9 = Extremely, and indicated how much they liked and disliked the game on a 9-point scale from 1 = Not at all to 9 = Very much. Cronbach's α for the continuous measure of enjoyment was at least .91 in all studies.

**Response time analyses**. All response time data (RT and RTSD) were log transformed to correct for skewness. In experiment 1, the tile game was not programmed with the intention of analyzing RT data. Specifically, it did not record RTs on trials where the tile was not activated (i.e. if the spacebar was pressed after

the gray tile disappeared, the RT was not recorded). It did, however, record RTs on trials where the tile was activated (i.e. RTs were recorded if the spacebar was pressed before the gray tile disappeared), so we analyzed these data in an exploratory fashion. The results were consistent with the findings from experiments 2 and 4: $I(M; E)$ had significant, negative effects on RT and RTSD (see Supplementary Information).

**Temporal difference prediction error and the value of information**. On each trial of the tile game, participants can choose to keep playing (i.e. they can continue the experiment) or quit (i.e. they can stop the experiment early and do something else). Let $A = \{play, quit\}$ be the set of possible actions. We assume that on each trial, $t \in \{1 \dots 50\}$, participants estimate the value (in dollars) of playing and quitting, denoted as $V_t(play)$ and $V_t(quit)$, respectively. If the decision is made to quit, all money earned up to that point is lost, so we let $V_t(quit)$ be the negative of the total earnings prior to trial $t$. If the decision is made to play, one of two states are observed: "hit" (if the tile is activated) or "miss" (if the tile is not activated). Let $m_t \in \{hit, miss\}$ be the state observed on trial $t$. The monetary value of $m_t$, denoted as $R(m_t)$, is known. In experiments 1, 2 and 4, $R(m_t = hit) = p_{E|M}(jackpot|hit) \times .1$ and $R(m_t = miss) = p_{E|M}(jackpot, |, miss) \times .1$. In experiment 3, $R(m_t = green) = .1$, $R(m_t = gray) = 0$, and $R(m_t = red) = -.02$. In experiment 5, $R(m_t = win) = .03$, $R(m_t = draw) = 0$, and $R(m_t = lose) = -.01$. Once $R(m_t)$ is observed, $V_t(play)$ is updated by means of temporal difference:

$$V_{t+1}(play) = V_t(play) + \alpha\delta(m_t) \tag{4}$$

where $\alpha \in [0, 1]$ is a learning rate, and $\delta(m_t)$ is the temporal difference prediction error elicited by the observation of $m_t$:

$$\delta(m_t) = R(m_t) - V_t(play) \tag{5}$$

Each time the decision is made to play, information is obtained, and this information can be used to increase future reward — that is, the information has value. Let $VOI_t(play)$ denote the value of the information associated with choosing to play on trial $t$. It is computed as follows:

$$VOI_t(play) = \mathbb{E}_{p(R(m_t)|play)}\left[\sum_{a \in A}[\pi_t'(a) - \pi_t(a)]F_t(a)\right] \tag{6}$$

$\pi_t(a)$ is the probability of choosing action $a \in A$ on the current trial $t$. It is the output of the softmax choice rule:

$$\pi_t(a) = \frac{e^{\beta V_t(a)}}{\sum_{i \in A} e^{\beta V_t(i)}} \tag{7}$$

where $\beta \in \mathbb{R}_{\geq 0}$ is an inverse temperature parameter that controls the explore-exploit tradeoff. $F_t(a)$ is what the new value of action $a$ would be if the choice was made to play on trial $t$:

$$F_t(a) = \begin{cases} V_t(a) + \alpha\delta(m_t) \text{ if } a = play \\ V_t(a) - R(m_t) \text{ if } a \neq play \end{cases} \tag{8}$$

$\pi_t'(a)$ is what the new probability of choosing action $a$ would be if the choice was made to play on trial $t$:

$$\pi_t'(a) = \frac{e^{\beta F_t(a)}}{\sum_{i \in A} e^{\beta F_t(i)}} \tag{9}$$

The expectation in equation 6 is taken under the probability distribution $p(R(m_t)|play)$, which is identical to $p_M(m)$ in each experiment. Intuitively, $VOI_t(play)$ quantifies the degree to which choosing to play on trial $t$ would improve the profitability of the participant's action policy (i.e. the participant's probability of choosing to play versus quit on the next trial).

We ran 1,000 simulations for each combination of $p_{E|M}(e, |, m)$ and $p_M(m)$, with $\alpha$ set to .9 and $\beta$ set to 5. For each simulation, we assumed that on all 50 trials, the choice was made to play, and computed (i) the average value of $VOI_t(play)$ and (ii) the average of the absolute value of $\delta(m_t)$. For both variables, we computed the average output of each simulation to get our final estimates of $VOI$ and $\delta(M)$.

**Observe condition**. In the observe condition of experiment 4, let $X$ denote whether the tile is activated or not, and let $I(X; E)$ denote the mutual information between $X$ and $E$. Our manipulation of $I(X; E)$ in the observe condition paralleled our manipulation of $I(M; E)$ in the play condition. We randomly selected the value of $p_X(activated)$ from the set $\{.2, .3, .4, .5, .6, .7, .8\}$ and the value of $p_{E|X}(jackpot|activated)$ from the set $\{.6, .7, .8, .9, 1\}$ with the constraint that neither parameter could be identical across the two games. We let $p_{E|X}(jackpot|not\ activated) = 1 - p_{E|X}(jackpot|activated)$. Thus, the only difference between the play condition and the observe condition is that, in the observe condition, tile-activation was not a means. To keep participants' eyes on the screen, we required that the spacebar be pressed at the end of each trial to advance to the next round.

**$I(M; E)$ for hand games**. In Rock, Paper, Scissors, $M$ can take on nine different values and $E$ can take on three different values. Each value of $M$ corresponds to a

possible combination of symbols selected by the player and their opponent: {rock-rock, rock-paper, rock-scissors, paper-rock, paper-paper, paper-scissors, scissors-rock, scissors-paper, scissors-scissors}. Each value of $E$ corresponds to an outcome: {win, lose, draw}. Because the opponent chooses randomly, $p_E(e) = 1/3$ for all values of $e$ regardless of the strategy participants employ (e.g., choosing hands at random versus always choosing the same hand). Therefore, $H(E)$ always equals 1.58, where $H(E)$ is Shannon entropy of $E$:

$$H(E) = -\sum_{e \in \mathscr{E}} p_E(e) \log_2 p_E(e) \tag{10}$$

The value of $E$ is fully determined by the value of $M$: $p_{E|M}(win|m) = 1$ when $m \in$ {rock-scissors, paper-rock, scissors-paper}, $p_{E|M}(draw|m) = 1$ when $m \in$ {rock-rock, paper-paper, scissors-scissors}, and $p_{E|M}(lose|m) = 1$ when $m \in$ {scissors-rock, rock-paper, paper-scissors}. Therefore, $H(E|M)$ always equals 0, where $H(E|M)$ is the Shannon entropy of $E$ conditional on $M$:

$$H(E|M) = -\sum_{e \in \mathscr{E}, m \in \mathscr{M}} p_{E,M}(e, m) \log_2 \frac{p_{E,M}(e, m)}{p_M(m)} \tag{11}$$

Subtracting $H(E|M)$ from $H(E)$ gives $I(M; E)$, so in Rock, Paper, Scissors, $I(M; E) = 1.58$.

In Odds vs. Evens, $M$ can take on four different values and $E$ can take on two different values. Each value of $M$ corresponds to a possible combination of numbers selected by the player and their opponent: {1–1, 1–2, 2–1, 2–2}. Each value of $E$ corresponds to an outcome: {win, lose}. The opponent chooses hands randomly, so $H(E) = 1$ regardless of the strategy participants employ. As in Rock, Paper, Scissors, the value of $E$ is fully determined by the value of $M$: For players who choose odds, $p_{E|M}(win|m) = 1$ when $m \in$ {1–2, 2–1}, and $p_{E|M}(lose|m) = 1$ when $m \in$ {1–1, 2–2}, and for players who choose evens, $p_{E|M}(win|m) = 1$ when $m \in$ {1–1, 2–2}, and $p_{E|M}(lose|m) = 1$ when $m \in$ {1–2, 2–1}. Therefore, $H(E|M) = 0$, and $I(M;E) = H(E) - H(E|M) = 1$.

**Reporting Summary**. Further information on research design is available in the Nature Research Reporting Summary linked to this article.

## Data availability

The data generated in this study have been deposited at OSF. Source data are provided with this paper.

## Code availability

Custom code used for data preprocessing, analysis, and simulation are available at OSF Custom code used for data collection are available at GitHub.

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

## Acknowledgements

This work was funded by the Yale Center for Customer Insights, the Yale School of Management Behavioral Research Lab, and the Automaticity of Cognition, Motivation, and Emotion Lab at Yale University. D.E.M. thanks Lisa Feldman Barrett and Dana Brooks for their mentorship and support.

## Author contributions

The theory was conceptualized by D.E.M. and subsequently refined by all authors. The experiments were conceptualized by all authors, and coded by D.E.M. Back-end code for data collection was developed by R.W.C. Analyses were performed by P.E.S. and D.E.M., figures were created by R.W.C. and D.E.M., and guidance on data interpretation and visualization was provided by all authors. The first draft of the manuscript was written by D.E.M., and subsequent writing and editing was performed by all authors.

## Competing interests

The authors declare no competing interests.
