## [Peer Review File · Nature Communications]

Title: A Computational Theory of the Subjective Experience of FlowREVIEWER COMMENTS

Reviewer #1 (Remarks to the Author):

This is among the most interesting and provocative papers I've read in a while. It stands out as an excellent candidate for publication in Nature Communications for several reasons. The topic is important; the methods are rigorous; it bridges several fields; and, it is highly novel and creative. I have some comments below, because critical feedback will of greatest help to the authors. But, my overall evaluation is that this is a very strong manuscript and should be published.

As I understand it, flow is understood to be a property of active engagement in a task (e.g., playing tennis or a video game, or driving a car). I am not aware that people generally describe attentional engagement in the absence of a task as "flow" (e.g., watching an absorbing movie, or watching flocks of birds make interesting patterns in the sky).

If this characterization is accurate, it feels somewhat at odds with the sparseness of the "mutual information" account of flow provided here. The essence of the mutual information account is that we experience flow when the information provided by "means" states increases our certainty about upcoming "end" states. It is not clear why this theory should be restricted to active tasks. The experiments presented in this manuscript all involve active tasks, but it is easy to imagine passive tasks with the same property. Suppose that participants were passively presented with M states leading to E states equivalent to those in Experiment 1. Here, the M states would be randomly selected by the experimenter with probability p_m . Would there be any less "flow" (i.e., attentional engagement and enjoyment) in this case? In this case, would "flow" also depend on mutual information between M and E?

Suppose that the answer is "yes" to both questions — basically, that nothing in the results of this experiment depends on the context of an "active" task at all. In that case, I feel like really the experiments and theory pertain to "focused, enjoyable attention" but not to flow as traditionally understood, since (as I understand it) flow is traditionally theorized as a state attained only during "active" tasks (e.g. tennis, not movie-watching). Perhaps this new theory of attention would inform our understanding of flow, but it wouldn't be a model of flow—certainly not a comprehensive one.

Suppose that the answer is "no" to both questions—that the set of results presented here really necessitate that participants have an active role in the task, and are not merely passive observers. In that case, it must be that something more than mutual information is involved, since mutual information can be a property of states that are merely passively observed. Perhaps the authors can make strong claims about what that "something extra" is, based on experimental evidence. More likely they cannot. Either way, the fact that "something extra" is required would be an important point to make. The mutual information piece that the authors focus on here would just be one part of a large, yet-to-be-elaborated model of flow.

Finally, a few more detailed notes. I found the transition from Experiment 2 to Experiment 3 confusing the first time I read it. Maybe it would be helpful to segregate discussion of Experiments 1 and 2 into two parts: "Alternative accounts we can rule out at this point" (marginal value, entropy and skill-challenge balance) and "alternative accounts we can't rule out, and address in experiment 3" (correlation and controllability). This might allow for a more seamless transition into Experiment 3.

Currently the introduction doesn't present alternative models to the author's proposed account. Later in the manuscript, of course, alternatives are considered and rejected (marginal value, skill-challenge balance, correlation, controllability). The overall narrative might be more compelling if these alternatives — or at least some of them — had been presented in the introduction.

I didn't find the paragraph at 472, on real-world applications, terribly convincing. It felt like these were just thin gestures in the direction of application. Maybe that's just a matter of taste. But, if this paragraph is to be maintained, I wonder it would be possible to spell out how exactly one might go about increasing mutual information, in one of these domains, in a more specific way. Then we might see whether the application leads to surprising and novel interventions, or instead in the kind of interventions that seem like they would be obviously helpful (but now with a more detailed computational understanding of the precise reasons why).

At 503 I didn't have a clear understanding of where one draws the line between approximating a thing and operating "as if" one were approximating it.

Reviewer #2 (Remarks to the Author):

Summary & Overall Impression

This paper proposes and tests a novel "informational theory of flow," which holds that the mutual information between ends (goals) and the means of attaining them determines flow. The authors demonstrate that the proposed mathematical definition correlates with subjective reports of flow in a stylized stochastic reward task. Overall, the paper is well-written.

Considered as a theoretical contribution, the core idea of the paper is intriguing and furnishes conceptual connections between psychology and work from other disciplines. The central mathematical concept of mutual information is simple, tractable, and makes a number of definite and unique predictions, meaning that the paper's central hypothesis is eminently testable. The paper's principal aim is also ambitious, as identifying the computational basis of flow would represent a significant advance in our understanding.

With these notes of praise in mind, I think the theoretical dimension of the paper requires more development to reach its full potential. The work is currently structured more like an empirical paper that tests an existing theory than a joint theory & empirical paper that proposes and explores a novel idea. Thus, elements that would make the theoretical exposition more compelling are at times clipped

or missing. For example, the paper discusses the proposal's conceptual antecedents, but does not fully draw out its implications. How does mutual information explain the presence (or absence) of flow in stylized situations such as playing a video game (or working a menial job)? More generally, how does the theory account for the principal findings of the huge body of existing work on the determinants of flow?

Empirically, the work is well-organized and communicated overall. However, the structural similarity between all three experiments, particularly the shared mechanism for manipulating $p(\text{hit})$, undercuts the paper's case for generality. The presentation of findings also obscures some aspects of the data: the findings are principally communicated in the form of parametric statistical tests, which only reveal the degree to which mutual information outcompetes other specific hypotheses, not the degree to which it fits the data in absolute terms. It remains somewhat unclear, for example, whether there are any systematic mispredictions that we should be aware of.

Critiques & Suggested Improvements

- I think the paper would benefit from more discussion of how the theory can be applied to situations other than the stylized experiment presented in the paper. This both fixes ideas and forces one to contend with the model's scope of applicability. For example, I can see how equation 1 applies to tasks with neatly defined goals and actions, such as chess, tennis, or competitive gambling; here, the model even seems to make the intuitive prediction that flow should increase with personal skill and attain at a maximum when one plays opponents that are at, or slight above, their own level (where making good strategic choices is necessary to win but the outcome is still far from certain). On the other hand, I am left wondering how the model accounts for the wide varieties of activities that occasion flow but cannot easily be operationalized using equation 1. Take, for instance, reading a good book. What are the "means" and "ends" in this case? Could we in principle measure the mutual information between them?

- The model seems to deviate from existing conceptions of flow in a number of ways. Does the theory predict that flow depends only on the probabilistic structure of reward contingency and not, e.g., on: (1) properties of the individual, such as their expectations or amount of relevant experience, (2) whether the rewards are changed to punishments or multiplied by several orders of magnitude, and (3) the nature of the environment were the task is performed? I think such predictions, especially ones that are surprising or contradict other models, should be discussed more thoroughly.

- The strength of the empirical work rests the fact that all three experiments are variations on the same basic structure. In particular, while the authors make a compelling case that their timing-based manipulation was successful in setting $p(\text{hit})$, their further claim that this represent a full-fledged operationalization of the underlying theory is, I think, subject to question; if the fundamental theory is about the relationship between means and ends, exogenously manipulating control over the means in a way that is opaque to subjects introduces a degree of artificiality into the design. A carefully measured source of natural variation should be sufficient to replicate the results in more diverse environments without this potential confound.

- Experiments 1/2 yield rich data about the relationship between flow and the two manipulated

probabilities, but this information is obscured by the reliance on parametric statistical tests. As illustrated in the left-hand pane of Figure 2, mutual information depends on these manipulated probabilities in a characteristic nonlinear way. This relationship is an extremely precise prediction of the hypothesis: average flow should be concave in $p(\text{hit})$ for each value of $p(\text{jackpot}|\text{hit})$, with the concavity and level increasing over successive values of $p(\text{jackpot}|\text{hit})$. A nonparametric analysis of this point could be included in a number of different ways, such as a table mirroring Figure 2 or a plot. A transparent view into the data will, I think, more precisely communicate how strong the results are.

- On a related point, from the stated regressions we only know that mutual information fits the data better than expected value and a few alternatives, but it is hard to tell how good the fit is in absolute terms. This problem is exacerbated by the fact that measures of fit, such as R-squared values, are omitted from the paper, leaving the reader to wonder how sizable mutual information's contribution to flow really is.

Final Impression & Recommendation

Overall, I think the core idea is promising and deserves further development and consideration. In particular, I would recommend that the authors develop the theoretical content of the paper and discuss how it fits with existing theory, explains existing evidence, and predicts flow across a variety of situations both intuitively and mathematically. I would also recommend additional empirical work that verifies the results in a way that does not rely on a similar manipulation of $p(\text{hit})$. Finally, I would suggest reworking the presentation of existing empirical results to reduce distance between the reader and the data.

Reviewer #3 (Remarks to the Author):

“Flow” is the subjective state of being immersed and engaged in one’s current task. The present study aims to characterize the computational processes underlying this subjective state by drawing on the notion of “mutual information” in information theory. The authors suggest that mutual information between desired end states (goals) and means of attaining them gives rise to flow.

The authors tested this hypothesis in 3 experiments implementing a computer-based task (“tile game”) which was designed to achieve precise control over each possible value of the manipulated variables (means and end states). At the end of each trial, measures of “flow” were reported by the participant using a Likert scale from 1 to 9, then regressed on the value of mutual information on each trial.

Experiment 2 assessed other subjective experiences related to flow such as task enjoyment and attention, while experiment 3 was designed to rule out the contribution of other possible alternatives, such as controllability, money incentives for task performance, associative strength, etc.

In all three experiments, the authors found that increasing mutual information increases the level of flow experienced by participants, speeds up response times, and makes the game more enjoyable.

This is a very interesting work at the intersection of various fields, including positive psychology, intrinsic

motivation and value-based decision-making. When working with subjective reports in reward-guided paradigms, it is always difficult to dissociate what is elaborated retrospectively from what is actually experienced, i.e. to dissociate a posthoc value effect from a true subjective experience. One strength of this work is to control “by design” this (retrospective) effect of value on subjective experience by having a condition where flow can increase following a punishment.

I have some general comments on the general positioning of the paper in the literature, as well as on the consistency of the results with other studies on flow or concepts related to flow (fluency) and on the relationship between flow and controllability. Also, some points in the Methods section are not always very clear and the analysis and interpretation of the results require some clarification:

1. First of all, a general remark. I think it should be discussed, or at least mentioned, that the notion of flow as operationalized in the present study applies to rather simple (and therefore necessarily simplified) actions followed by rather unambiguous feedback. However, most of our behaviours/actions are sequential and hierarchically organized, with sometimes ambiguous effects as to the particular action in the sequence that caused them. This would help connect with traditional description of “flow” in the literature, as a state that results from well-learned, skilled tasks involving sequences of continuous actions for which particular outcomes simply cannot be computed or monitored – such as a concert performer playing the piano (Csikszentmihalyi, 2000, but the same applies to professional athletes reporting similar experiences, see Lafon, 2012). As the authors suggest, monitoring the statistical dependency (or a rough approximation of it) between action and outcome is certainly possible, but may be particularly difficult for these behaviours that involve sequential actions followed by multiple outcomes occurring at the same time.

2. A related point: fluent selection and execution of actions (“sense of mastery”, “knowing what to do and how to do it”), rather than increased dependency between action and outcome, is often reported to describe flow experiences among experts. Fluent selection would be the consequence of reduced competition and conflict between programmes for selected and alternative actions (see e.g. Wenke et al., 2010, Cisek, 2007, Nachev et al., 2005). I wonder to what extent it is possible to reconcile this view (the “selection” view) with the “informational” view of flow.

Indeed, in a sense, the informational view is an “ideomotor” view of the flow experience, in that it focuses on the link between action and outcome while ignoring the uncertainty associated with the (downstream) processes of selection of the action itself (Chambon & Haggard, 2013). But in real-world, the way in which actions are selected (or even prepared) has an impact on how the action itself and any subsequent outcomes are experienced, above and beyond their mutual dependency. Hence the question: would a participant experience maximal flow in a situation where $I(M;E)$ is also maximal but there’s ambiguity in which target to select to get the desired outcome? This echoes the debated notion of “success through chance” in action theory (Greco, 2009), where a poorly skilled archer achieves the desired outcome (winning a prize) by the right means (hitting the bullseye) – and thus $I(M;E)$ is maximal – but does not feel in control or even responsible for the outcome because it was not intentional (means selection was not appropriate) (see Chambon & Haggard, 2013 for a discussion of this example).

3. On page 2, the characterization of the flow experience in terms of “means-ends fusion” is quite

fascinating in that it makes a striking echo to the phenomenon of "intentional binding". Intentional binding is a widely-reported compression of the perceived time between an action and the resulting outcome (also sometimes called "action-outcome fusion"), and is considered an implicit measure of the human sense of agency (Haggard, Clark, & Kalogeras, 2002). Just as the self-reported flow increases with increasing I(M;E), intentional binding is greater when the action predicts the subsequent outcome, i.e. action-effect fusion is maximal when prediction error is minimal (for a review, see Moore & Obhi, 2012). Beyond the fact that I would be very interested to know the extent to which mutual information predicts the strength of intentional binding (IB) in your task (which should be the case, given what we know about the link between IB and outcome predictability), I also suspect that IB, as an *implicit* measure of subjective control, will be better predicted by mutual information than the explicit measure of perceived controllability, such as the one you use in your study. Interestingly, intentional binding is known to be greater in trials (t) following an error (at t-1), for a reason that could be motivational (see Di Costa et al., 2018). Did you look at such inter-trial effects, for example by adding the previous trials (success/failure) as a factor in your linear model? Is self-reported flow experience higher on trials following an error/failure?

4. I very much appreciated that the authors control the effect of controllability on self-reported flow, as perceived controllability (or sense of agency, sense of control, etc.) is sometimes also described in terms of mutual information. The Liljeholm's group, for example, uses a highly-related information measure, the Jensen-Shannon (JS) divergence, to quantify controllability. Specifically, JS divergence quantifies the divergence between outcome probability distributions for all actions available at a given time, and has been shown to be associated with increased control or sense of control over action outcomes (eg Liljeholm et al., 2013, Mistry & Liljeholm, 2016; see also Chambon et al., 2018 for a similar operationalization). The results reported in this work are therefore at odds with previous results suggesting that perceived controllability is a function of the statistical dependency (or mutual information, or any related quantity) between action and outcome. I think this should be discussed (even briefly) in the text.

5. Data analysis. There is a lack of information on the regression model used to estimate the contribution of each factor to self-reported flow. In Exp2, it is unclear whether you fitted a model including all the predictors shown (i.e., a linear combination of Enjoy, Choice, RT, RTSD, etc., what Figure 3 seems to suggest) or whether you performed independent linear regressions for each predictor? Figure 3 is not clear on this point: "flow" is presented with other predictors (Enjoy, Choice, RT...), but I guess that flow is the dependent variable, not a predictor. More generally, what I would have expected is a table of parameter estimates (as in Figure 3) with, on the vertical axis, the list of predictors included in your regression model (e.g. flow=MI+Enjoy+Choice+RT...). Also, it is not clear why you did not test models with interaction terms – it is not absurd to consider that some of your predictors may interact with each other.

6. Data analysis. It is not clear to me how the effect of I(M;E) on flow was adjusted for EV: was EV added in the regression model as an independent predictor? Would it be possible to have a table of parameter estimates for the model including EV (or any reward-related quantity varying on a trial-by-trial basis) as

an independent predictor? What if you replace EV by a binary variable (1 for success, 0 for failure/punishment)?

Is there a reason why you used monetary incentives rather than e.g. smileys to signal success in the task? Wouldn't the use of smileys have made it easier to rule out any value effect on self-reported flow? (even though social reinforcement-learning experiments suggest that a smile can be model as a scalar reward)

7. The results of Experiment 3 are impressive and convincing. But I think that the authors cannot get rid of the “value” effects found in Experiments 1 and 2 so quickly. These effects are real, and the inconsistency between these effects and those from Experiment 3 should be further discussed and understood. What the data from the 3 experiments *taken together* show is that self-reported flow has many contributors. The key contributor is certainly mutual information, but a significant part of the variance in self-reported flow can be attributed to EV or ΔV in exp1 and 2. Figure 5 shows that in exp3 a condition is associated with a greater flow experience on average, but again, this does not rule out that some variance in self-reported flow is attributable to monetary incentives experienced on a trial-by-trial basis.

I would recommend to better discuss these inconsistencies in the results of the 3 experiments (e.g., why value explains part of the self-reported flow in the first two experiments, but in experiment 3, flow experience is greater in conditions with penalties) and to qualify some assertions in the text about the lack of effect of monetary incentives on self-reported flow, e.g.: “monetary incentives had no independent effect on flow regardless of whether they were operationalized as the expected value of task performance or the marginal value of successful versus unsuccessful task performance”.

8. Minor: was there an incentive not to pressing the space bar as soon as possible? Or a penalty for early presses? I ask the question because the optimal strategy in this task should be to press the space bar continuously so that you never miss the response time window.

References

- Csikszentmihalyi, M. (2000). *Beyond boredom and anxiety* (25th Anniversary ed.). San Francisco: Jossey-Bass.
- Lafont, D. (2015). *Back to the Zone: Sport and Inner Experience*. Breakaway Books.
- Cisek, P. (2007). Cortical mechanisms of action selection: The affordance competition hypothesis. *Philosophical Transactions of the Royal Society B*, 362, 1585–1599
- Nachev, P., Rees, G., Parton, A., Kennard, C., & Husain, M. (2005). Volition and conflict in human medial frontal cortex. *Current Biology*, 15, 122–128.
- Wenke, D., Fleming, S. M., & Haggard, P. (2010). Subliminal priming of actions influences sense of control over effects of action. *Cognition*, 115(1), 26-38.
- Chambon, V., & Haggard, P. (2013). Premotor or Ideomotor: How Does the Experience of Action Come About?. *Action science: Foundations of an emerging discipline*, 359.
- Greco, J. (2009). Knowledge and success from ability. *Philosophical Studies*, 142, 17–26.
- Haggard, P., Clark, S., & Kalogeras, J. (2002). Voluntary action and conscious awareness. *Nature*

neuroscience, 5(4), 382-385.

Moore, J. W., & Obhi, S. S. (2012). Intentional binding and the sense of agency: a review. *Consciousness and cognition*, 21(1), 546-561.

Di Costa, S., Théro, H., Chambon, V., & Haggard, P. (2018). Try and try again: Post-error boost of an implicit measure of agency. *Quarterly Journal of Experimental Psychology*, 71(7), 1584-1595.

Liljeholm, M., Wang, S., Zhang, J., & O'Doherty, J. P. (2013). Neural correlates of the divergence of instrumental probability distributions. *Journal of Neuroscience*, 33(30), 12519-12527.

Mistry, P., & Liljeholm, M. (2016). Instrumental divergence and the value of control. *Scientific reports*, 6(1), 1-10.

Chambon, V., Thero, H., Findling, C., & Koehlin, E. (2018). Believing in one's power: a counterfactual heuristic for goal-directed control. *bioRxiv*, 498675.

Reviewer #4 (Remarks to the Author):

The study formulated a computational theory of flow, utilizing the concept of goal fusion and mutual information. The study gave empirical evidence supporting the model. I really liked the motivation of the authors to provide a computational account of flow, and the topic is quite timely given the recent growing body of computational literature on intrinsic motivation and curiosity.

However, I feel that the model that the authors provided is too simplistic or narrow, and I suspect that the model is valid only for a specific type of task in a limited parameter space. Below I would like to elaborate this point by providing some examples.

First, the model lacks (or is indifferent to) one critical component of flow --- feeling of agency or self-determination. The task they used is actually pretty effective to induce strong task engagement because participants believe that their effort can determine (although illusory) the outcome of means. But think about a task in which the outcome of means is simply determined by a computer or lottery and participants know it. Many decision making tasks have that feature. Even if this lottery task has the same P_M and $P_{E|M}$ with the current task, we can easily imagine that the lottery task is much more boring. But the current model cannot distinguish these two scenarios because they have the identical P_M and $P_{E|M}$. Note that, if feeling of controllability is assessed, the lottery task should show much lower feeling of controllability (the definition of control is a matter for discussion in the literature, e.g., Huys & Dayan, 2009, but I just talk about subjective feelings here), so feeling of controllability would be a better predictor of flow than mutual information.

Second, with the current model, if the outcome of means is perfectly informative of the end outcome (i.e. $P_{E|M} = 0$ or 1), the model predicts that the task is most interesting when the success rate is 50%

(with a binary outcome). But the relationship between success rate and enjoyment is a matter of controversy (see also Dubey & Griffith, 2020). There is good suggestion that the relationship is inverted-U shaped, but I am not sure if the past studies uniformly suggested that 50% is the "sweet spot". My sense is that this sweet spot is dependent on nature of the task (e.g., feeling of cost).

Third, it was great to see that the authors tested a new prediction from the model. But again, I wondered whether the results are dependent on the task. In the current version of the task, the activation of a tile would lead to the outcome of 10, 0, and -2 with the probabilities of 50%, 25%, and 25%. But think about a new task in which the probabilities are 25%, 50%, and 25%. Here P_M has the same entropy (thus the mutual information is unchanged) but I can imagine that this task is quite demotivating because there is only 25% chance of getting the highest reward. The critical issue is that, in our real-life decision making, even if there are multiple consequences, we normally have a clear single goal in mind (in this case, to get +10) and the behavior is regulated and motivated in relation to the goal. But the current model does not concern whether the consequence is goal relevant or not. All consequences are evaluated only in terms of their probabilities.

I must admit that these are my strong guesses (and I might be wrong!), but the point is that the current empirical data are not sufficient to address these various important factors that have been implicated in the literature of flow or intrinsic motivation.

I also wonder if it was a good idea to use goal fusion as the framework to present the model. Goal system theory typically concerns the congruence between the outcome of means and the end outcome ($P_{\{E|M\}}$). In other words, it does not concern the entropy of means (P_M). Means and ends are congruent when you decide to eat healthy foods (means) to achieve the goal of losing weights (ends). But this theoretical framework does not usually take into account the probability of successfully eating healthy foods (P_M), and if it does by any chance, I suspect it would predict that intrinsic motivation is highest when the mean has high probability of success, not when participants are uncertain about their capability to eat healthy foods. Goal system theory is quite useful for making a prediction when people can successfully self-regulate behavior but the proposed model does not seem to be compatible with goal system theory in this context, which further makes me wonder about the generalizability of the proposed model.

Minor comments:

Controllability and skill-challenge balance were examined separately from the $I(M; E)$ to predict flow. But I think it is important to include all of these predictors altogether in a single regression model to tease apart the independent contribution of these variables. That being said, I have a feeling that controllability is also part of the omnibus concept of flow and I wonder if it makes sense to pit the current model against controllability and skill-challenge balance. For example, we can consider that skill-challenge balance is represented in the entropy of P_M in the context of the current task.

There are some deviations from the preregistration, which is noted in the SOM. I think it is important to clearly state that these decisions were made *prior to* data analysis, to ensure that their changes are not the consequence of data exploration. Or it is best to simply report the preregistered results in the first place.

It was not clear how the authors addressed the nested structure of the data in the analysis as each participant has two data points.

I really would like to see a scatter plot of I (M; E) and flow. Are they really linearly related? Can we see large individual differences? Effect size alone does not seem to be sufficient to evaluate the validity of the model.

Kou Murayama (I always sign after 2020)

Summary of Revisions

We thank the editor and referees for their invaluable feedback on our manuscript, “A Computational Theory of Flow” (NCOMMS-20-49113-1). We have completed a major revision of our manuscript, which includes two new preregistered experiments, additional analyses, and expansions to our introduction and discussion based on reviewer comments. Despite these additions, we have remained within the 5,000-word limit. Below, we provide a brief overview of our additions and changes, followed by point-by-point replies.

New Experiments. Our new experiments address the major theoretical questions raised by the reviewers, including: (1) Is our theory compatible with the fact that flow is restricted to tasks that involve active engagement? (R1.2, R3.3, R4.2), (2) Does our theory generalize to tasks other than the one we used in our original experiments? (R2.2, R2.4, R2.6, R4.4), (3) Does our theory generalize to games of chance, where $I(M;E)$ can be high while agency and self-determination are low? (R3.3, R4.2), and (4) If an increase in $I(M;E)$ coincides with a decrease in success rate, does flow still increase as our theory predicts? (R4.4).

To address question 1, we ran a new experiment (experiment 4 in our revised manuscript) in which we randomly assigned participants to a “play condition,” where participants actively played the tile game (identical to experiments 1 and 2), or an “observe condition,” where participants merely observed the game unfold. We found that the mutual information between tile-activation and jackpot-winning predicted flow in the play condition, but not the observe condition, suggesting that our effects rely on participants having an active role in the task. Critically, this is exactly what our theory predicts should happen: In our revised manuscript, we clarify that, according to our theory, it is not mutual information *in general* that predicts flow, but mutual information *between M and E, specifically*. Indeed, the key difference between the observe condition and the play condition is that, in the observe condition, there is no M, because for observers, tile-activation is not a means. A means, as we define it, is a state one brings about to achieve their goal, and in the “observe condition,” participants play no role in bringing about the state of the tile being activated (or not activated). Thus, in the “observation condition,” tile-activation is merely an event that signals the probability of a jackpot, rather than a means of goal attainment. By showing that mutual information predicts flow if, and only if, the mutual information is between M and E, we address a major question raised by multiple reviewers, while providing additional support for our model and replicating the results of experiments 1 and 2.

To address questions 2-4, we ran a new experiment (experiment 5) in which participants played two of the world’s oldest and most enduring games: Rock, Paper, Scissors (which dates back to the Chinese Han dynasty) and Odds vs. Evens (which dates back to ancient Rome). In Rock, Paper, Scissors, $I(M;E)=1.58$, and in Odds vs. Evens, $I(M;E)=1$, so our theory predicts that Rock, Paper, Scissors elicits more flow, and this is what we found. Critically, both games are games of chance — you cannot get more reward by trying harder or paying more attention, leaving no room for agency or self-determination. Nevertheless, our theory correctly identified which game elicits more flow, and the level of flow in each game was on par with amount of flow elicited by versions of the tile game with equivalent values of $I(M;E)$, suggesting that, when $I(M;E)$ is held constant, reducing agency does not reduce flow. Our results also suggest that when an increase in $I(M;E)$ coincides with a decreases in success rate, flow goes up, not down, just as our theory predicts. Indeed, in Rock, Paper, Scissors, the probability of winning is $1/3$, and in Odds vs. Evens, the probability of winning

is 1/2. Yet we confirmed our preregistered prediction that Rock, Paper, Scissors (which has a higher value of $I(M;E)$) elicits more flow.

In sum, our new experiments (1) confirm our theory's prediction that flow is related to mutual information only in environments that involve active task engagement (i.e. environments with means and ends), (2) generalize our theory to novel activities, (3) generalize our theory to games of chance, and (4) confirm our theory's prediction that when an increase in $I(M;E)$ coincides with a decrease in success rate, flow goes up.

New Analyses. Following the recommendations of Reviewer 2, we have taken many steps to give readers a complete picture of our data, and to illustrate the degree to which our data fit our model's predictions in absolute terms. First, we have supplemented our linear regressions with generalized additive models (GAMs) — a statistical technique in which outcomes are assumed to depend on smooth, nonparametric functions of the predictors. Unlike linear regression, GAMs can discover nonlinearities that would violate the informational theory of flow. What we found, however, supports our theory: the effect of $I(M;E)$ on flow is everywhere positive (Fig. 2B in main text). We also used GAMs to estimate the degree to which the data fit our predictions in absolute terms. Specifically, we used a GAM to model flow in terms of $p(\text{hit})$ and $p(\text{jackpot}|\text{hit})$, and generated a matrix containing the predicted value of flow for each combination of the two parameters (Fig. 2C in main text). If our theory is correct, this matrix should align with the matrix representing $I(M;E)$ in terms of $p(\text{hit})$ and $p(\text{jackpot}|\text{hit})$ (Fig 1B). Consistent with this, the two matrices were correlated at $r = .88$. In addition to these extra analyses, we have added effect sizes (standardized betas) for all regression results, and have clarified exactly which variables were included as outcomes and predictors in each analysis.

Following the recommendation of Reviewer 4 (R4.7) we have changed our statistical reporting slightly so that the analyses in the main text are more faithful to our preregistered exclusion plans, and our analyses based on deviations from preregistration (deviations we decided on prior to data analysis) are provided in the supplement. Both sets of analyses yield the same results, in terms of what is significant and what is not, but reviewers will notice slight changes in the exact numbers.

We have made some unsolicited additions that we feel strengthen our paper. Specifically, we have pit $I(M;E)$ against two additional constructs: temporal difference prediction errors (from computational reinforcement learning models) and the “value of information.” (see Page 12 and experiments 3 and 5). We show that $I(M;E)$ outperforms both of these constructs in predicting flow.

In summary, our new analyses (1) confirm that the function mapping from $I(M;E)$ to flow is monotonically increasing, (2) confirm that our model fits our data well in absolute terms, (3) clarify the structure of our statistical analyses, and (4) demonstrate that our model outperforms models of flow as a positive function of VOI or TD error.

Reviewer #1

R1.1: This is among the most interesting and provocative papers I've read in a while. It stands out as an excellent candidate for publication in Nature Communications for several reasons. The topic is important; the methods are rigorous; it bridges several fields; and, it is highly novel and creative. I have some comments

below, because critical feedback will of greatest help to the authors. But, my overall evaluation is that this is a very strong manuscript and should be published.

We thank Reviewer 1 for their positive comments and helpful feedback. We have addressed Reviewer 1's suggestions by clarifying aspects of the manuscript and running additional experiments, which we describe below.

R1.2: As I understand it, flow is understood to be a property of active engagement in a task (e.g., playing tennis or a video game, or driving a car). I am not aware that people generally describe attentional engagement in the absence of a task as “flow” (e.g., watching an absorbing movie, or watching flocks of birds make interesting patterns in the sky). If this characterization is accurate, it feels somewhat at odds with the sparseness of the “mutual information” account of flow provided here. The essence of the mutual information account is that we experience flow when the information provided by “means” states increases our certainty about upcoming “end” states. It is not clear why this theory should be restricted to active tasks. The experiments presented in this manuscript all involve active tasks, but it is easy to imagine passive tasks with the same property. Suppose that participants were passively presented with M states leading to E states equivalent to those in Experiment 1. Here, the M states would be randomly selected by the experimenter with probability p_m . Would there be any less “flow” (i.e., attentional engagement and enjoyment) in this case? In this case, would “flow” also depend on mutual information between M and E? Suppose that the answer is “yes” to both questions — basically, that nothing in the results of this experiment depends on the context of an “active” task at all. In that case, I feel like really the experiments and theory pertain to “focused, enjoyable attention” but not to flow as traditionally understood, since (as I understand it) flow is traditionally theorized as a state attained only during “active” tasks (e.g. tennis, not movie-watching). Perhaps this new theory of attention would inform our understanding of flow, but it wouldn't be a model of flow—certainly not a comprehensive one. Suppose that the answer is “no” to both questions—that the set of results presented here really necessitate that participants have an active role in the task, and are not merely passive observers. In that case, it must be that something more than mutual information is involved, since mutual information can be a property of states that are merely passively observed. Perhaps the authors can make strong claims about what that “something extra” is, based on experimental evidence. More likely they cannot. Either way, the fact that “something extra” is required would be an important point to make. The mutual information piece that the authors focus on here would just be one part of a large, yet-to-be-elaborated model of flow.

We agree with Reviewer 1 that it is important to establish whether our findings replicate in a “passive” task, where participants do not have an active role in bringing about the end state. Accordingly, we ran the suggested experiment (Experiment 4 in our revised manuscript): We randomly assigned participants to a “play condition,” where participants actively played the tile game (identical to experiments 1 and 2), or an “observe condition,” where participants merely observed the game unfold. Our experimental design ensured that participants in both conditions kept their eyes on the game the entire time (i.e., participants in the “observe condition” could not ignore the events on the screen). We found that the mutual information between tile-activation and jackpot-winning predicted flow in the play condition, but not the observe condition, suggesting that our effects rely on participants having an active role in the task. We agree with Reviewer 1 that these results indicate that something more than mutual information is involved in generating flow. In our revised manuscript, we specify that, according to our theory, it is not mutual information *in general* that predicts flow, but mutual information *between M and E, specifically*. Indeed, the key difference between the observe condition and the play condition is that, in the observe condition, there is no M, because for observers, tile-activation is not a *means*. A means, as we define it, is a state one brings about to achieve their goal, and in the “observe condition,” participants play no role in bringing about the state of the tile being activated (or not activated). Thus, in the “observation condition,”

tile-activation is merely an event that signals the probability of a jackpot, rather than a means of goal attainment.

R1.3: Finally, a few more detailed notes. I found the transition from Experiment 2 to Experiment 3 confusing the first time I read it. Maybe it would be helpful to segregate discussion of Experiments 1 and 2 into two parts: “Alternative accounts we can rule out at this point” (marginal value, entropy and skill-challenge balance) and “alternative accounts we can’t rule out, and address in experiment 3” (correlation and controllability). This might allow for a more seamless transition into Experiment 3.

We thank Reviewer 1 for this suggestion. We have moved our discussion of marginal value, entropy, and skill-challenge balance to subsections under experiments 1 and 2 (lines 215-244), leaving the yet-to-be-ruled-out variables for experiment 3 (lines 246-284).

R1.4: Currently the introduction doesn’t present alternative models to the author’s proposed account. Later in the manuscript, of course, alternatives are considered and rejected (marginal value, skill-challenge balance, correlation, controllability). The overall narrative might be more compelling if these alternatives — or at least some of them — had been presented in the introduction.

We agree with this suggestion and have added a paragraph to the end of our introduction previewing three of the alternative variables that we pit our theory against: skill-challenge balance, controllability, and expected value (lines 105-111).

R1.5: I didn’t find the paragraph at 472, on real-world applications, terribly convincing. It felt like these were just thin gestures in the direction of application. Maybe that’s just a matter of taste. But, if this paragraph is to be maintained, I wonder it would be possible to spell out how exactly one might go about increasing mutual information, in one of these domains, in a more specific way. Then we might see whether the application leads to surprising and novel interventions, or instead in the kind of interventions that seem like they would be obviously helpful (but now with a more detailed computational understanding of the precise reasons why).

To stay within the 5,000-word limit, we opted to remove our speculations on real-world implications and focus instead on fleshing out ideas about the adaptive function of the link between $I(M;E)$ and flow — specifically, the possibility that this link helps people resolve the explore-exploit dilemma (lines 340-353).

R1.6: At 503 I didn’t have a clear understanding of where one draws the line between approximating a thing and operating “as if” one were approximating it.

We agree that approximating a thing and operating “as if” one were approximating it may be a distinction without a difference. We have removed this sentence.

Reviewer #2

R2.1: This paper proposes and tests a novel “informational theory of flow,” which holds that the mutual information between ends (goals) and the means of attaining them determines flow. The authors demonstrate that the proposed mathematical definition correlates with subjective reports of flow in a stylized stochastic reward task. Overall, the paper is well-written.

Considered as a theoretical contribution, the core idea of the paper is intriguing and furnishes conceptual connections between psychology and work from other disciplines. The central mathematical concept of mutual information is simple, tractable, and makes a number of definite and unique predictions, meaning that

the paper's central hypothesis is eminently testable. The paper's principal aim is also ambitious, as identifying the computational basis of flow would represent a significant advance in our understanding.

With these notes of praise in mind, I think the theoretical dimension of the paper requires more development to reach its full potential. The work is currently structured more like an empirical paper that tests an existing theory than a joint theory & empirical paper that proposes and explores a novel idea. Thus, elements that would make the theoretical exposition more compelling are at times clipped or missing. For example, the paper discusses the proposal's conceptual antecedents, but does not fully draw out its implications. How does mutual information explain the presence (or absence) of flow in stylized situations such as playing a video game (or working a menial job)? More generally, how does the theory account for the principal findings of the huge body of existing work on the determinants of flow?

We thank Reviewer 2 for their positive comments, and for pushing us to develop our theory further. We significantly expanded our theoretical exposition while adhering to our 5,000-word limit. First, we followed Reviewer 2's recommendation to illustrate how our theory explains the presence and absence of flow in stylized situations. On lines 95-104, we illustrate how our theory explains the ability of slot machines to induce high levels of flow. We also show that our theory correctly predicts how a slot machine could be altered to elicit less flow. In addition to our slot machine example, we highlight the ways in which our theory disagrees with existing frameworks that attribute flow to skill-challenge balance and controllability.

R2.2: Empirically, the work is well-organized and communicated overall. However, the structural similarity between all three experiments, particularly the shared mechanism for manipulating $p(\text{hit})$, undercuts the paper's case for generality.

We agree with Reviewer 2 that it is important to generalize our findings beyond the basic paradigm used in our first three experiments. Accordingly, we ran a new experiment (experiment 5) that uses completely different tasks. Specifically, we computed the value of $I(M;E)$ for two of the world's oldest and most enduring games: Rock, Paper, Scissors (which dates back to the Chinese Han dynasty) and Odds vs. Evens (which dates back to ancient Rome). In Rock, Paper, Scissors, $I(M;E)=1.58$, and in Odds vs. Evens, $I(M;E)=1$, so our theory predicts that Rock, Paper, Scissors elicits more flow. We obtained clear support for this prediction, which we preregistered. In addition to generalizing our findings to a new task, experiment 5 shows that our theory passes the critical test of making correct predictions about existing tasks developed without any intention of supporting our theory.

R2.3: The presentation of findings also obscures some aspects of the data: the findings are principally communicated in the form of parametric statistical tests, which only reveal the degree to which mutual information outcompetes other specific hypotheses, not the degree to which it fits the data in absolute terms. It remains somewhat unclear, for example, whether there are any systematic mispredictions that we should be aware of.

We have taken many steps to give readers a complete picture of our data. First, we have supplemented our linear regressions with generalized additive models (GAMs) — a statistical technique in which outcomes are assumed to depend on smooth, nonparametric functions of the predictors. Unlike linear regression, GAMs can discover nonlinearities that would violate the informational theory of flow. What we found, however, supports our theory: the effect of $I(M;E)$ on flow is everywhere positive (Fig. 2B in main text). We also used GAMs to estimate the degree to which the data fit our predictions in absolute terms. Specifically, we used a GAM to model flow in terms of $p(\text{hit})$ and $p(\text{jackpot} | \text{hit})$, and generated a matrix containing the predicted value of flow for

each combination of the two parameters (Fig. 2C in main text). If our theory is correct, this matrix should align with the matrix representing $I(M;E)$ in terms of $p(\text{hit})$ and $p(\text{jackpot}|\text{hit})$ (Fig 1B). Consistent with this, the two matrices were correlated at $r = .88$. In addition to these extra analyses, we have added effect sizes (standardized betas) for all regression results, and have clarified exactly which variables were included as outcomes and predictors in each analysis.

R2.4: Critiques & Suggested Improvements

- I think the paper would benefit from more discussion of how the theory can be applied to situations other than the stylized experiment presented in the paper. This both fixes ideas and forces one to contend with the model's scope of applicability. For example, I can see how equation 1 applies to tasks with neatly defined goals and actions, such as chess, tennis, or competitive gambling; here, the model even seems to make the intuitive prediction that flow should increase with personal skill and attain at a maximum when one plays opponents that are at, or slight above, their own level (where making good strategic choices is necessary to win but the outcome is still far from certain). On the other hand, I am left wondering how the model accounts for the wide varieties of activities that occasion flow but cannot easily be operationalized using equation 1. Take, for instance, reading a good book. What are the "means" and "ends" in this case? Could we in principle measure the mutual information between them?

We completely agree that our model's scope of applicability warrants further discussion, and that, in many activities, it is far from obvious what the "means" is and what the "end" is. Reading a book is a good example. We do believe, however, that any goal-directed activity can in principle be broken down into means and ends. On lines 66-71 of our revised manuscript, we have shown how two activities lacking obvious means and ends can be broken down this way. One is Reviewer 2's example of reading a good book. We show that for someone reading a book, E could denote whether or not the protagonist's fate is discovered, and M could denote whether or not the next chapter is finished. Of course, there are many other ways to represent the task of reading a book — our example simply shows that, in principle, the task can be broken down into a means and end. Our second example is the activity of dancing a tango: E could denote whether the dance partner is impressed, and M could denote whether the right foot steps forward passing the left foot. Immediately after these examples, we defend our claim that any goal-directed activity can be broken down into a means and end by noting that the definition of "goal-directed activity" entails the existence of a means (the activity) and an end (the goal to which the activity is directed). Moreover, on lines 95-104, we added a third, extended illustration of our theory's scope of applicability: We show that our theory can explain why flow is elicited by slot machines. Finally, we conducted a new experiment (experiment 5, lines 313-331) that tests our theory in a completely different experimental context: the hand games, Rock, Paper, Scissors and Odds vs. Evens. These are two of the world's oldest and most enduring games. Rock, Paper, Scissors dates back to the Chinese Han dynasty and Odds vs. Evens dates back to ancient Rome. In Rock, Paper, Scissors, $I(M;E)=1.58$, and in Odds vs. Evens, $I(M;E)=1$, so our theory predicts that Rock, Paper, Scissors elicits more flow. We obtained clear support for this prediction, which we preregistered. This experiment, along with our expanded introduction, lends support to our assumption that our theory applies to a broad range of activities.

R2.5: The model seems to deviate from existing conceptions of flow in a number of ways. Does the theory predict that flow depends only on the probabilistic structure of reward contingency and not, e.g., on: (1) properties of the individual, such as their expectations or amount of relevant experience, (2) whether the rewards are changed to punishments or multiplied by several orders of magnitude, and (3) the nature of the environment where the task is performed? I think such predictions, especially ones that are surprising or contradict other models, should be discussed more thoroughly.

We also agree that these points are worth discussing further. On page 4 of our revised manuscript (lines 105-111), we highlight how the informational theory of flow diverges from other models. For instance, we note that our theory sometimes says flow should be relatively high when skill-challenge balance and controllability are relatively low. We also note that our theory assumes that flow is insensitive to variation in instrumental value, allowing for flow to persist, and even grow stronger, in the face of diminishing rewards and increasing punishments.

On lines 253-258, we show that the results of our third experiment contradict predictions derived from two basic psychological principles: (i) organisms aim to avoid punishment (this suggests that flow should be greatest in our “neutral condition,” which involves zero punishment), and (ii) negative stimuli automatically capture attention (this suggests that flow should be greatest in our “punishment condition,” where negative stimuli are most frequent). Contrary to these principles, our theory predicts that flow should be highest in the “mixture condition,” where punishment is neither most frequent or least frequent. In our revised manuscript, we use this unique prediction to highlight how our theory diverges from alternative frameworks.

Our theory does not deny that flow depends on properties of the individual or the surrounding environment, and we have clarified this in our revised manuscript. Our theory allows the surrounding environment to play a role to the extent that the surrounding environment can influence the value of $I(M;E)$. This happens, for instance, to basketball players shooting free throws in front of the home crowd versus an away crowd. $p(M = \text{make shot})$ is lower in the away environment versus the home environment, which influences the mutual information between M (make vs. miss) and E (earn point vs. don't earn point). Regarding the influence of properties of the individual, we clarify that, according to our theory, flow depends on how people represent their tasks (how they construe their means and ends), and people's beliefs about the relevant probability distributions (e.g., beliefs about the probability of success) — factors that vary from individual to individual.

R2.6: The strength of the empirical work rests the fact that all three experiments are variations on the same basic structure. In particular, while the authors make a compelling case that their timing-based manipulation was successful in setting $p(\text{hit})$, their further claim that this represent a full-fledged operationalization of the underlying theory is, I think, subject to question; if the fundamental theory is about the relationship between means and ends, exogenously manipulating control over the means in a way that is opaque to subjects introduces a degree of artificiality into the design. A carefully measured source of natural variation should be sufficient to replicate the results in more diverse environments without this potential confound.

In our new experiment 5, the manipulation of $I(M;E)$ does not rely on the timing-based manipulation from experiments 1-4. As Reviewer 2 suggested, we leveraged natural variation in $I(M;E)$ across two similar, but slightly different activities. Thus, the results of experiment 5 establish that our findings generalize to situations less artificial than the one we devised in experiments 1-4.

R2.7: Experiments 1/2 yield rich data about the relationship between flow and the two manipulated probabilities, but this information is obscured by the reliance on parametric statistical tests. As illustrated in the left-hand pane of Figure 2, mutual information depends on these manipulated probabilities in a characteristic nonlinear way. This relationship is an extremely precise prediction of the hypothesis: average flow should be concave in $p(\text{hit})$ for each value of $p(\text{jackpot} | \text{hit})$, with the concavity and level increasing over successive values of $p(\text{jackpot} | \text{hit})$. A nonparametric analysis of this point could be included in a number of different ways, such as a table mirroring Figure 2 or a plot. A transparent view into the data will, I think, more precisely communicate how strong the results are.

On a related point, from the stated regressions we only know that mutual information fits the data better than expected value and a few alternatives, but it is hard to tell how good the fit is in absolute terms. This problem is exacerbated by the fact that measures of fit, such as R-squared values, are omitted from the paper, leaving the reader to wonder how sizable mutual information's contribution to flow really is.

As discussed above, we have added new analyses and visualizations to give readers a more transparent view of the data. Our additions include GAMs, effect sizes, and a comparison of how flow and $I(M;E)$ relate to $p(m)$ and $p(e|m)$. The latter analysis serves as an estimate of how well our model fits the data in absolute terms. As mentioned above, the empirically observed values of flow at each level of $p(m)$ and $p(e|m)$ correlated with our model's predictions at $r = .88$. We opted for this approach over R-squared values because we have no principled way of establishing an R-squared benchmark (i.e. how high the R-squared would be if our theory were true). Ideally, 100% of the variance in self-reports of flow would be explained by three factors: (1) the true value of $I(M;E)$, (2) random variation in people's estimates of $I(M;E)$ (estimates of $I(M;E)$ are noisy reflections of the true value), and (3) random variation in how people map their subjective experiences of flow onto response scales. However, factors 2 and 3 are unknown, so we do not know how much variance should be explained by factor 1. Our alternative approach addresses this problem. It averages out the variance attributable to factors 2 and 3 above by taking the mean value of flow for each combination of $p(m)$ and $p(e|m)$, and correlating these values with the true value of $I(M;E)$. The correlation between these matrices gives a relatively pure estimate of the correspondence between flow and the true value of $I(M;E)$.

R2.8: Final Impression & Recommendation

Overall, I think the core idea is promising and deserves further development and consideration. In particular, I would recommend that the authors develop the theoretical content of the paper and discuss how it fits with existing theory, explains existing evidence, and predicts flow across a variety of situations both intuitively and mathematically. I would also recommend additional empirical work that verifies the results in a way that does not rely on a similar manipulation of $p(\text{hit})$. Finally, I would suggest reworking the presentation of existing empirical results to reduce distance between the reader and the data.

Thank you for these great suggestions for further developing our work theoretically and empirically—we believe addressing them has greatly improved our manuscript.

Reviewer #3

R3.1: “Flow” is the subjective state of being immersed and engaged in one's current task. The present study aims to characterize the computational processes underlying this subjective state by drawing on the notion of “mutual information” in information theory. The authors suggest that mutual information between desired end states (goals) and means of attaining them gives rise to flow.

The authors tested this hypothesis in 3 experiments implementing a computer-based task (“tile game”) which was designed to achieve precise control over each possible value of the manipulated variables (means and end states). At the end of each trial, measures of “flow” were reported by the participant using a Likert scale from 1 to 9, then regressed on the value of mutual information on each trial. Experiment 2 assessed other subjective experiences related to flow such as task enjoyment and attention, while experiment 3 was designed to rule out the contribution of other possible alternatives, such as controllability, money incentives for task performance, associative strength, etc.

In all three experiments, the authors found that increasing mutual information increases the level of flow experienced by participants, speeds up response times, and makes the game more enjoyable.

This is a very interesting work at the intersection of various fields, including positive psychology, intrinsic motivation and value-based decision-making. When working with subjective reports in reward-guided paradigms, it is always difficult to dissociate what is elaborated retrospectively from what is actually experienced, i.e. to dissociate a posthoc value effect from a true subjective experience. One strength of this work is to control “by design” this (retrospective) effect of value on subjective experience by having a condition where flow can increase following a punishment.

Thank you!

R3.2: I have some general comments on the general positioning of the paper in the literature, as well as on the consistency of the results with other studies on flow or concepts related to flow (fluency) and on the relationship between flow and controllability. Also, some points in the Methods section are not always very clear and the analysis and interpretation of the results require some clarification:

First of all, a general remark. I think it should be discussed, or at least mentioned, that the notion of flow as operationalized in the present study applies to rather simple (and therefore necessarily simplified) actions followed by rather unambiguous feedback. However, most of our behaviours/actions are sequential and hierarchically organized, with sometimes ambiguous effects as to the particular action in the sequence that caused them. This would help connect with traditional description of “flow” in the literature, as a state that results from well-learned, skilled tasks involving sequences of continuous actions for which particular outcomes simply cannot be computed or monitored – such as a concert performer playing the piano (Csikszentmihalyi, 2000, but the same applies to professional athletes reporting similar experiences, see Lafon, 2012). As the authors suggest, monitoring the statistical dependency (or a rough approximation of it) between action and outcome is certainly possible, but may be particularly difficult for these behaviours that involve sequential actions followed by multiple outcomes occurring at the same time.

We completely agree that the sequential and hierarchical nature of many activities poses significant computational challenges. Understanding how humans overcome these challenges is one of our primary goals going forward as we continue to develop our theory. Accordingly, on page 20, we now highlight the fact that in many activities (but not ours), actions are sequential and hierarchically organized, posing substantial computational challenges.

R3.3: A related point: fluent selection and execution of actions (“sense of mastery”, “knowing what to do and how to do it”), rather than increased dependency between action and outcome, is often reported to describe flow experiences among experts. Fluent selection would be the consequence of reduced competition and conflict between programmes for selected and alternative actions (see e.g. Wenke et al., 2010, Cisek, 2007, Nachev et al., 2005). I wonder to what extent it is possible to reconcile this view (the “selection” view) with the “informational” view of flow. Indeed, in a sense, the informational view is an “ideomotor” view of the flow experience, in that it focuses on the link between action and outcome while ignoring the uncertainty associated with the (downstream) processes of selection of the action itself (Chambon & Haggard, 2013). But in real-world, the way in which actions are selected (or even prepared) has an impact on how the action itself and any subsequent outcomes are experienced, above and beyond their mutual dependency. Hence the question: would a participant experience maximal flow in a situation where $I(M;E)$ is also maximal but there’s ambiguity in which target to select to get the desired outcome? This echoes the debated notion of “success through chance” in action theory (Greco, 2009), where a poorly skilled archer achieves the desired outcome (winning a prize) by the right means (hitting the bullseye) – and thus $I(M;E)$ is maximal – but does not feel in control or even responsible for the outcome because it was not intentional (means selection was not appropriate) (see Chambon & Haggard, 2013 for a discussion of this example).

The question Reviewer 3 raises is an important one, and we addressed it by running a new experiment (experiment 5). In this experiment, we tested our theory using games of chance: Rock, Paper, Scissors and Odds vs. Evens. In Rock, Paper, Scissors, $I(M;E)=1.58$, and in Odds vs. Evens, $I(M;E)=1$, so our theory predicts that Rock, Paper, Scissors elicits more flow, and this is what we found. Critically, both games are games of pure chance — you cannot get more reward by trying harder or paying more attention. And it satisfies Reviewer 3’s requirement that the task involve ambiguity about which target to select (all successes are “successes through chance”). Despite these deviations from the tile game, we again obtained support for our theory. Experiment 5 includes a discussion of these topics and cites Chambon & Haggard, 2012 and Chambon & Haggard, 2013.

R3.4: On page 2, the characterization of the flow experience in terms of “means-ends fusion” is quite fascinating in that it makes a striking echo to the phenomenon of “intentional binding”. Intentional binding is a widely-reported compression of the perceived time between an action and the resulting outcome (also sometimes called “action-outcome fusion”), and is considered an implicit measure of the human sense of agency (Haggard, Clark, & Kalogeras, 2002). Just as the self-reported flow increases with increasing $I(M;E)$, intentional binding is greater when the action predicts the subsequent outcome, i.e. action-effect fusion is maximal when prediction error is minimal (for a review, see Moore & Obhi, 2012). Beyond the fact that I would be very interested to know the extent to which mutual information predicts the strength of intentional binding (IB) in your task (which should be the case, given what we know about the link between IB and outcome predictability), I also suspect that IB, as an *implicit* measure of subjective control, will be better predicted by mutual information than the explicit measure of perceived controllability, such as the one you use in your study. Interestingly, intentional binding is known to be greater in trials (t) following an error (at $t-1$), for a reason that could be motivational (see Di Costa et al., 2018). Did you look at such inter-trial effects, for example by adding the previous trials (success/failure) as a factor in your linear model? Is self-reported flow experience higher on trials following an error/failure?

The connection between means-end fusion and intentional binding is fascinating. This is definitely something we will look to pursue in future work. Unfortunately, our data do not allow us to run the sort of fine-grained analyses Reviewer 3 suggests. Since we did not measure flow on a trial-by-trial basis, we cannot explore the possibility that flow is higher on trials following errors/failure. We apologize that this important methodological detail was not fully clear from our initial submission. We have revised our description of our flow measure to make this point clear for readers. That said, we look forward to developing a measure of flow that gives us greater temporal resolution so that we can see how flow unfolds throughout the task, and test questions like this one.

R3.5: I very much appreciated that the authors control the effect of controllability on self-reported flow, as perceived controllability (or sense of agency, sense of control, etc.) is sometimes also described in terms of mutual information. The Liljeholm’s group, for example, uses a highly-related information measure, the Jensen-Shannon (JS) divergence, to quantify controllability. Specifically, JS divergence quantifies the divergence between outcome probability distributions for all actions available at a given time, and has been shown to be associated with increased control or sense of control over action outcomes (eg Liljeholm et al., 2013, Mistry & Liljeholm, 2016; see also Chambon et al., 2018 for a similar operationalization). The results reported in this work are therefore at odds with previous results suggesting that perceived controllability is a function of the statistical dependency (or mutual information, or any related quantity) between action and outcome. I think this should be discussed (even briefly) in the text.

This is a great catch. We hadn’t considered how our controllability findings jibe with the “instrumental divergence” view of controllability, but we agree that there is some tension. On Page

13 (lines 240-244) of our revised manuscript, we discuss Liljeholm et al.'s work connecting controllability to JS divergence, and how our data challenges this perspective.

R3.6: Data analysis. There is a lack of information on the regression model used to estimate the contribution of each factor to self-reported flow. In Exp2, it is unclear whether you fitted a model including all the predictors shown (i.e., a linear combination of Enjoy, Choice, RT, RTSD, etc., what Figure 3 seems to suggest) or whether you performed independent linear regressions for each predictor? Figure 3 is not clear on this point: “flow” is presented with other predictors (Enjoy, Choice, RT...), but I guess that flow is the dependent variable, not a predictor. More generally, what I would have expected is a table of parameter estimates (as in Figure 3) with, on the vertical axis, the list of predictors included in your regression model (e.g. $\text{flow} = \text{MI} + \text{Enjoy} + \text{Choice} + \text{RT} \dots$). Also, it is not clear why you did not test models with interaction terms – it is not absurd to consider that some of your predictors may interact with each other.

We thank Reviewer 3 for this suggestion. We completely revamped the way we present the results to make absolutely clear which variables were included in each model, and whether we treated them as outcomes or predictors. We now follow Reviewer 3's suggestion of providing tables listing the outcomes and predictors of each model (Figures 2, 3, 4, 5, and 8). Our reason for omitting interaction terms is that we had no *a priori* predictions about interactions, and, relatedly, did not ensure that we would have sufficient statistical power to detect interactions. With over 100 possible interaction terms, we worried about increasing our risk of Type I error. However, we have made all of our data and code publicly available, enabling anyone to explore our data for possible interaction effects. Of course, if Reviewer 3 has any specific interactions in mind, we would be happy to run them and report the results.

R3.7: Data analysis. It is not clear to me how the effect of $I(M;E)$ on flow was adjusted for EV: was EV added in the regression model as an independent predictor? Would it be possible to have a table of parameter estimates for the model including EV (or any reward-related quantity varying on a trial-by-trial basis) as an independent predictor? What if you replace EV by a binary variable (1 for success, 0 for failure/punishment)? Is there a reason why you used monetary incentives rather than e.g. smileys to signal success in the task? Wouldn't the use of smileys have made it easier to rule out any value effect on self-reported flow? (even though social reinforcement-learning experiments suggest that a smile can be model as a scalar reward)

Thanks for raising this point. To adjust for EV, we regressed flow on EV and $I(M;E)$ simultaneously. We have clarified this in the revised manuscript, and, in general, have clarified the structure of each statistical model through updated figures. Also, our figure now displays the parameter estimates corresponding to the effect of EV (along with all other reward-related quantities) on every single outcome.

Regarding the possibility of replacing EV with a binary variable, we believe this is not possible (although we may have misunderstood the suggestion). The binary variable would indicate, on each trial, whether a jackpot was received or not. However, since flow was not measured on a trial-by-trial basis, this precludes the possibility of regressing flow (measured once per game) on the proposed trial-level binary variable. As noted in our reply to R3.4, we have revised our methods to help clarify how frequently we measured flow, and hope this detail will now be clear to readers.

Regarding the question of money versus positive/negative imagery, we opted for money because, as Reviewer 3 suggests, positive and negative imagery can be modeled as scalar rewards, but unlike money, their values are difficult to quantify. We reasoned that by using monetary incentives, we could establish with greater confidence whether effects of $I(M;E)$ on flow are attributable to reward.

Further, one strength of using real money is to demonstrate the comparative power of increasing $I(M;E)$ compared to increasing monetary incentives, the latter of which is often considered the gold standard (at least by behavioral economists) for increasing engagement in a given task.

R3.8: The results of Experiment 3 are impressive and convincing. But I think that the authors cannot get rid of the “value” effects found in Experiments 1 and 2 so quickly. These effects are real, and the inconsistency between these effects and those from Experiment 3 should be further discussed and understood. What the data from the 3 experiments *taken together* show is that self-reported flow has many contributors. The key contributor is certainly mutual information, but a significant part of the variance in self-reported flow can be attributed to EV or ΔV in exp1 and 2. Figure 5 shows that in exp3 a condition is associated with a greater flow experience on average, but again, this does not rule out that some variance in self-reported flow is attributable to monetary incentives experienced on a trial-by-trial basis. I would recommend to better discuss these inconsistencies in the results of the 3 experiments (e.g., why value explains part of the self-reported flow in the first two experiments, but in experiment 3, flow experience is greater in conditions with penalties) and to qualify some assertions in the text about the lack of effect of monetary incentives on self-reported flow, e.g.: “monetary incentives had no independent effect on flow regardless of whether they were operationalized as the expected value of task performance or the marginal value of successful versus unsuccessful task performance”.

We thank Reviewer 3 for encouraging us to discuss the results of experiment 3 in more detail. In our view, the reward-related variables were correlated with flow in experiments 1 and 2 because all of these variables shared $I(M;E)$ as a common cause. For instance, ΔV was correlated with flow not because ΔV had a causal effect on flow, but because both variables were caused by $I(M;E)$. This would explain why the correlations between flow and reward-related variables vanished in experiment 3; in this experiment, we managed to disentangle the reward-related variables from $I(M;E)$, eliminating the common cause that induced the original correlations. However, as Reviewer 3 suggests, this is not the only possible explanation. It could be the case that one or more of the reward-related variables has a causal effect on flow, but the effect of $I(M;E)$ on flow is so much stronger that it wipes out the influence of the others (this would happen in experiment 3, but not experiments 1 and 2, because it is in experiment 3 that $I(M;E)$ cuts against the reward-related variables. Accordingly, we have removed all statements implying that we ruled out the possibility that reward-related variables contribute to flow. We limit ourselves to the more modest claim that we ruled out the possibility that reward-related variables account for our effects of $I(M;E)$.

R3.9: Minor: was there an incentive not to pressing the space bar as soon as possible? Or a penalty for early presses? I ask the question because the optimal strategy in this task should be to press the space bar continuously so that you never miss the response time window.

Yes, we disincentivized pressing the space bar as fast as possible by displaying a warning message for 3.5 seconds (which feels quite long within the task) whenever participants responded preemptively. We have clarified this on page 6 of our revised manuscript.

Reviewer #4

R4.1: The study formulated a computational theory of flow, utilizing the concept of goal fusion and mutual information. The study gave empirical evidence supporting the model. I really liked the motivation of the authors to provide a computational account of flow, and the topic is quite timely given the recent growing body of computational literature on intrinsic motivation and curiosity.

We thank Reviewer 4 for these positive comments.

R4.2: However, I feel that the model that the authors provided is too simplistic or narrow, and I suspect that the model is valid only for a specific type of task in a limited parameter space. Below I would like to elaborate this point by providing some examples.

First, the model lacks (or is indifferent to) one critical component of flow --- feeling of agency or self-determination. The task they used is actually pretty effective to induce strong task engagement because participants believe that their effort can determine (although illusory) the outcome of means. But think about a task in which the outcome of means is simply determined by a computer or lottery and participants know it. Many decision making tasks have that feature. Even if this lottery task has the same P_M and $P_{\{E|M\}}$ with the current task, we can easily imagine that the lottery task is much more boring. But the current model cannot distinguish these two scenarios because they have the identical P_M and $P_{\{E|M\}}$. Note that, if feeling of controllability is assessed, the lottery task should show much lower feeling of controllability (the definition of control is a matter for discussion in the literature, e.g., Huys & Dayan, 2009, but I just talk about subjective feelings here), so feeling of controllability would be a better predictor of flow than mutual information.

This is an excellent point. We completely agree that it is critical to establish that our theory applies in situations lacking in agency and self-determination. We have addressed this issue by expanding our theoretical discussion and adding a new experiment along the lines of what Reviewer 4 suggests. First, on page 4 (lines 95-104), we note that slot machines are notorious for inducing extreme levels of flow to the point that some players end up losing their homes and loved ones. We then show how our theory can account for this: slot machines have very high levels of $I(M;E)$ and if their $I(M;E)$ were reduced, they'd no longer develop such an unbreakable hold on those who play. This exposition demonstrates that, though counterintuitive, flow can be high when self-determination is low, and our theory is well equipped to explain why.

One of our new experiments (experiment 5) shows empirically that our theory applies to the low-agency situations that Reviewer 4 describes. We computed the value of $I(M;E)$ for two games of chance: Rock, Paper, Scissors and Odds vs. Evens. In both games, the outcome of our choice (e.g., "rock") entirely depends on a random event: the opponent's decision (e.g., "rock," "paper," or "scissors"). $I(M;E)$ is higher for Rock, Paper, Scissors, so we preregistered the prediction that this game would elicit more flow. Our experiment confirmed this prediction. Moreover, the level of flow in each game was on par with the level of flow in the tile game with equivalent values of $I(M;E)$, suggesting that, when $I(M;E)$ is held constant, reducing agency does not reduce flow.

These findings notwithstanding, we completely agree with Reviewer 4 that, inevitably, our theory will need to be developed further to account for findings it does not predict in its current form. Ultimately, this may include the finding that agency or self-determination plays an important role in the flow-generating process. But we believe that our new findings demonstrate that our theory can provide a solid foundation for what will be a long-term effort to achieve a formal understanding of flow.

R4.3: Second, with the current model, if the outcome of means is perfectly informative of the end outcome (i.e. $P_{\{E|M\}} = 0$ or 1), the model predicts that the task is most interesting when the success rate is 50% (with a binary outcome). But the relationship between success rate and enjoyment is a matter of controversy (see also Dubey & Griffith, 2020). There is good suggestion that the relationship is inverted-U shaped, but I am not sure if the past studies uniformly suggested that 50% is the "sweet spot". My sense is that this sweet spot is dependent on nature of the task (e.g., feeling of cost).

We completely agree that the relationship between success rate and flow depends on the nature of the task. Indeed, this insight anticipates an exciting aspect of our theory that we plan to pursue in future work: the informational theory of flow makes precise predictions about how task structure can completely alter the relationship between success rate and flow. Consider a slightly different version of the tile game, where the goal is not to earn a jackpot on each trial, but to earn as many jackpots in a row as possible — that is, to obtain the longest possible streak of jackpots. In this case, E denotes a non-negative integer corresponding to the length of the streak (instead of 0 or 1 corresponding to winning/not winning a jackpot on the current trial), and $p(e)$ is a geometric distribution (instead of a Bernoulli distribution). In this “streak” version of the tile game, it can be shown that $I(M;E)$ increases as the probability of success on a given trial approaches 1 (but goes to zero if when the probability of success reaches 100%). The reason for this is that, if E is geometrically distributed, $H(E)$ increases as the probability of success on a given trial approaches (but does not reach) 100%, which means more uncertainty can be reduced by observing M . Incidentally, a “streak” version of the tile game doesn’t just change the relationship between success rate and $I(M;E)$, it also results in a substantial increase in $I(M;E)$ in absolute terms, which could explain why so many app companies use streaks to keep people engaged. We are excited to present tests of these ideas as part of a follow-up paper using our theory to understand how different parameterizations of the same activity can shape the dynamics of flow. We are confident that this work will confirm Reviewer 4’s intuition that the “sweet spot” for success rates depends on the nature of the task, while lending further support to our theory.

R4.4: Third, it was great to see that the authors tested a new prediction from the model. But again, I wondered whether the results are dependent on the task. In the current version of the task, the activation of a tile would lead to the outcome of 10, 0, and -2 with the probabilities of 50%, 25%, and 25%. But think about a new task in which the probabilities are 25%, 50%, and 25%. Here P_M has the same entropy (thus the mutual information is unchanged) but I can imagine that this task is quite demotivating because there is only 25% chance of getting the highest reward. The critical issue is that, in our real-life decision making, even if there are multiple consequences, we normally have a clear single goal in mind (in this case, to get +10) and the behavior is regulated and motivated in relation to the goal. But the current model does not concern whether the consequence is goal relevant or not. All consequences are evaluated only in terms of their probabilities.

Reviewer 4 rightly points out that in experiment 3, our theory predicts that flow would remain the same if the probabilities were changed to $p(+10) = .25$, $p(+0) = .5$, $p(-2) = .25$. This is assuming, of course, that after the change, participants still construe E as having three possible values: +10, 0, and -2. We suspect they would, since people naturally distinguish between gains, non-gains, and losses (e.g., Tversky & Kahneman, 1992). However, if participants lumped non-gains and losses together, our theory would make a different prediction that aligns with Reviewer 4’s intuition. If participants construed E as a binary variable denoting +10 or “not +10,” the probabilities proposed by Reviewer 4 would reduce $I(M;E)$, so we would expect a reduction in flow. This highlights one of Reviewer 4’s important insights: flow depends on how people construe their task. Objectively, there may be three possible outcomes, but someone may only care about two possible outcomes: attaining vs. not attaining their goal. Again, this is extremely unlikely when the three outcomes are “gain,” “non-gain,” and “loss,” but as Reviewer 4 suggests, most activities can be construed in different ways depending on the relevance of each outcome to one’s current goal. In our view, one of the strengths of our theory is that it accounts for the fact that people may construe their activities in different ways, and makes predictions about how these construals alter flow. In our revised

manuscript, we highlight this feature on lines 60-71 by illustrating that a fundamental feature of the M and E in $I(M;E)$ is their perceiver-dependence.

Returning to Reviewer 4's thought experiment: Assuming that participants construe E as having three possible outcomes, what would happen if the probability of success, but not $I(M;E)$, were reduced? Our new experiment 5 allowed us to test this question. Participants played two games: Rock, Paper, Scissors and Odds vs. Evens. In Rock, Paper, Scissors, the probability of winning is $1/3$, and in Odds vs. Evens, the probability of winning is $1/2$. Conversely, $I(M;E)$ is higher in Rock, Paper, Scissors than in Odds vs. Evens. Yet we confirmed our preregistered prediction that Rock, Paper, Scissors elicits more flow than Odds vs. Evens.

We wish to zero in on one of Reviewer 4's comments in particular, because it suggests that our original manuscript was unclear in places. Specifically, Reviewer 4 says that "...the current model does not concern whether the consequence is goal relevant or not. All consequences are evaluated only in terms of their probabilities." Strictly speaking, our model *does* concern whether consequences are goal-relevant or not in the sense that we propose that flow is a positive function of M and E specifically, not mutual information in general. M and E are goal-relevant by definition. The implication is that in an activity where mutual information is high, but the mutual information is not between the goal-relevant variables M and E, flow will be low. We tested this prediction in a new preregistered experiment (experiment 4): We randomly assigned participants to a "play condition," where participants played the tile game, or an "observe condition," where participants merely observed the game unfold. In the "observe condition," tile-activation is not a means, because participants play no role in bringing about the state of the tile being active or disappearing. Thus, we predicted that the mutual information between tile-activation jackpot-winning would predict flow in the "active condition" (where tile-activation is a means) but not in the "observe condition" (where tile-activation is not a means). We confirmed this prediction, highlighting the fact that our theory is concerned not only with probabilities, but also with the goal-relevant nature of the variables.

R4.5: I must admit that these are my strong guesses (and I might be wrong!), but the point is that the current empirical data are not sufficient to address these various important factors that have been implicated in the literature of flow or intrinsic motivation.

I also wonder if it was a good idea to use goal fusion as the framework to present the model. Goal system theory typically concerns the congruence between the outcome of means and the end outcome ($P_{\{E|M\}}$). In other words, it does not concern the entropy of means (P_M). Means and ends are congruent when you decide to eat healthy foods (means) to achieve the goal of losing weights (ends). But this theoretical framework does not usually take into account the probability of successfully eating healthy foods (P_M), and if it does by any chance, I suspect it would predict that intrinsic motivation is highest when the mean has high probability of success, not when participants are uncertain about their capability to eat healthy foods. Goal system theory is quite useful for making a prediction when people can successfully self-regulate behavior but the proposed model does not seem to be compatible with goal system theory in this context, which further makes me wonder about the generalizability of the proposed model.

We completely agree with Reviewer 4 that $I(M;E)$ may deviate somewhat from the psychological concept of means-end fusion. In our revised manuscript, we clarify that although means-end fusion theory was part of the inspiration for our model of flow, $I(M;E)$ is not the only way to formalize the means-end fusion concept. In particular, on line 49, we clarify that many formulations of means-end fusion are possible.

R4.6: Minor comments:

Controllability and skill-challenge balance were examined separately from the I (M; E) to predict flow. But I think it is important to include all of these predictors altogether in a single regression model to tease apart the independent contribution of these variables. That being said, I have a feeling that controllability is also part of the omnibus concept of flow and I wonder if it makes sense to pit the current model against controllability and skill-challenge balance. For example, we can consider that skill-challenge balance is represented in the entropy of P_M in the context of the current task.

We appreciate this suggestion. We opted not to take the statistical approach suggested here due to the fact that the causal relationships between flow, controllability, and skill-challenge balance are ambiguous. As Reviewer 4 notes, controllability and skill-challenge balance could be outcomes of flow (e.g., experiencing flow could lead people to appraise their activity as high in skill-challenge balance and controllability) or they could be correlated with flow via a third variable (e.g., all three could be outcomes of I(M;E)). Thus, it would be very difficult to interpret the results of a multiple regression that includes all three variables as predictors.

R4.7: There are some deviations from the preregistration, which is noted in the SOM. I think it is important to clearly state that these decisions were made *prior to* data analysis, to ensure that their changes are not the consequence of data exploration. Or it is best to simply report the preregistered results in the first place.

We thank Reviewer 4 for raising this concern. Indeed, all deviations were chosen prior to data analysis. We have clarified this in our revised manuscript (lines 396-397), and mention these deviations earlier (they now appear in the second subsection of the methods, right after the “participants” section).

R4.8: It was not clear how the authors addressed the nested structure of the data in the analysis as each participant has two data points.

On lines 378-380 we have clarified that we used linear mixed effects models with subject-level random intercepts to account for the nested structure of our data.

R4.9: I really would like to see a scatter plot of I (M; E) and flow. Are they really linearly related? Can we see large individual differences? Effect size alone does not seem to be sufficient to evaluate the validity of the model.

As Reviewer 4 suggests, linear regressions alone do not allow us to detect nonlinearities that would violate our theory. Although our theory doesn't predict that the effect of I(M;E) on flow is strictly linear, it does predict that the effect is monotonically increasing — increasing I(M;E) increases flow over of the full range of I(M;E). We formally tested this prediction in our revised manuscript by fitting generalized additive models (GAMs) — a statistical technique in which outcomes are assumed to depend on smooth, nonparametric functions of the predictors. Unlike linear regression, GAMs can discover nonlinearities that would violate the informational theory of flow. What we found, however, supports our theory: the effect of I(M;E) on flow is everywhere positive (Fig. 2B in main text). The figures we created do not include scatter plots, because, to provide a clear view of the shape of the function (so readers can see the derivative is never negative), we had to truncate the y-axis, which would crop out a lot of data points. However, if Reviewer 4 prefers a “zoomed out” version of the plot that includes the individual data points, we are happy to modify accordingly.

REVIEWER COMMENTS

Reviewer #1 (Remarks to the Author):

I remain enthusiastic about the contributions of this manuscript and recommend publication. The revised version is much more readable, and the new experiments are valuable.

I have one lingering conceptual concern which the authors could address in a minor revision without peer review, or simply use to guide their future studies of this phenomenon. The key issue is whether we should think of the “means” as an action or as a state.

Especially in the introduction and the general discussion the authors encourage us to think of the “means” as an action. To me, this is a very natural interpretation of the word “means” in ordinary English. It also seems to capture a key idea in previous treatments of flow, which emphasize the fusion of action and outcome. Finally, it seems important to the connection with “empowerment” in AI (“the maximum of the mutual information between agent’s actions and end states”, line 42).

But, in the formal model and in all of the experiments, the “means” is a state. Moreover, in all the experiments, pains are taken to sever any relationship between action and subsequent states. (I assume the mutual information would be zero between, for instance, response time and reward in the original experiments, or between rock, paper, scissors actions and reward in the final game). Is the model nevertheless meant to capture the subjective feeling of a fusion between action and outcome? Is the link to “empowerment” still relevant? Is it important that the experiments seem to be designed to maintain the subjective impression that actions are causally efficacious (i.e., influence rewards), even though in fact they do not?

Reviewer #2 (Remarks to the Author):

Overall Impression

The revised manuscript addresses many of my comments and, from what I can tell, those of other reviewers. In particular, I appreciated the revamped data analysis, which I feel makes a clearer and more compelling case for the "computational theory of flow." Experiment 6 also provides additional evidence for a relationship between flow and mutual information across two real-life games. Overall, the authors convincingly demonstrate the core claim of the paper: that the mutual information between means and ends can be an important predictor of flow beyond alternative candidate measures based on the task's probabilistic structure (e.g. expected value) or self-report constructs (e.g. controllability). This is especially salient in experiments 3 and 5, where alternative measures are shown to make the wrong directional prediction about which of two tasks will generate more flow.

Remaining Critiques

In my opinion, a truly compelling empirical case for the theory would still require demonstrating that the specific non-linear functional relationship between task probabilities and flow predicted by mutual information holds up across a variety of contexts. This would determine whether mutual information itself drives flow, or whether mutual information merely correlates with the actual driver. Showing that mutual information and flow correlate across different games (as in experiment 6) is important, but ultimately does not address this more nuanced question. I will admit, however, that this is a heavy lift probably best left to future papers.

I also appreciated that the authors added further discussion of how the model might apply to real-life situations, such as reading a book or dancing the tango, but still came away feeling that the arguments presented here were a bit tenuous. It seems plausible, for instance, that someone might be uncertain about their ability to pull off a particular tango move, but I find it hard to imagine situations where someone is genuinely uncertain about their ability to continue reading a book, watching a movie, or listening to a gripping story. In its current form, therefore, I do not see how the theory can account for the very obvious but important fact that passive consumption of information seems to generate much of the flow people experience day to day. A similar point holds for environmental factors that do not change an activity's probabilistic contingencies. Perhaps the conclusion is simply that "means-ends fusion" is not the only determinant of flow; if this is the case, however, I think the paper would benefit from a more open and honest discussion of this point and its implications for our understanding of flow.

Finally, the authors also mention that "Most activities can be represented in multiple ways, making M and E perceiver-dependent," and indeed it strikes me that for many tasks means and ends can be redefined arbitrarily such that the mutual information between them is either large, small, or non-existent. This is important because it means that the theory's predictions are entirely driven by assumptions about how people carve up the world into means and ends. As mentioned above, this is not at all obvious for many of the everyday things we do, meaning that different researchers can come to dramatically different conclusions about what the theory says. A theory is not complete unless there is a principled way of nailing down these researcher degrees of freedom. Is there a way of eliciting what people themselves define "means" to be? Hypothetically, is there a way to measure the probabilistic contingencies necessary to calculate the theory's prediction in real-world activities? Etc.

Final Impression & Recommendation

The above points of critique aside, I am enthusiastic about the creative and provocative claim at the core of this paper. Personally, I think the paper in its current form makes a significant scientific contribution worth sharing with other researchers. I also think, however, that a paper which seeks to revolutionize our understanding of a concept is most scientifically useful if it lays out exactly how it can be applied to situations beyond its author's own experiments. In particular, the present paper would be greatly strengthened if it fully "closed the theory" by clarifying exactly how it could be operationalized in non-lab settings (particularly, by addressing the fraction problem of defining and measuring an activity's "means"). I would also recommend space be given for a more direct discussion of how (or whether) the

theory can account for stylized features of flow that do not neatly fit into the a means-end framework, e.g. that passive information consumption can generate flow.

Reviewer #3 (Remarks to the Author):

The authors have addressed all my concerns and comments. I recommend publication.

Reviewer 1

R1.1. I remain enthusiastic about the contributions of this manuscript and recommend publication. The revised version is much more readable, and the new experiments are valuable.

We thank Reviewer 1 for these positive comments and helpful feedback on our previous draft.

R1.2. I have one lingering conceptual concern which the authors could address in a minor revision without peer review, or simply use to guide their future studies of this phenomenon. The key issue is whether we should think of the “means” as an action or as a state. Especially in the introduction and the general discussion the authors encourage us to think of the “means” as an action. To me, this is a very natural interpretation of the word “means” in ordinary English. It also seems to capture a key idea in previous treatments of flow, which emphasize the fusion of action and outcome. Finally, it seems important to the connection with “empowerment” in AI (“the maximum of the mutual information between agent’s actions and end states”, line 42). But, in the formal model and in all of the experiments, the “means” is a state. Moreover, in all the experiments, pains are taken to sever any relationship between action and subsequent states. (I assume the mutual information would be zero between, for instance, response time and reward in the original experiments, or between rock, paper, scissors actions and reward in the final game).

We agree that our definition of "means" was left somewhat ambiguous and have clarified our definition in our revision. We define "means" as states evoked by goal-directed motor commands — not motor commands *per se*. In our revised manuscript, our definition appears on line 60 ("*M* denotes a state brought about to achieve a goal") and is clarified further on lines 65-66: "Also note that, as this example illustrates, *M* denotes a state brought about by a goal-directed motor command (e.g., hitting or missing a bullseye), not the motor command itself (e.g., the motor command that implement dart throwing)." This revision emphasizes the fact that, despite being states rather than motor commands, means are *coupled* with (consequences of) motor commands.

R1.3. Is the model nevertheless meant to capture the subjective feeling of a fusion between action and outcome? Is the link to “empowerment” still relevant? Is it important that the experiments seem to be designed to maintain the subjective impression that actions are causally efficacious (i.e., influence rewards), even though in fact they do not?

We believe that our definition of "means" (a state elicited by a goal-directed motor command, rather than the motor command itself) is consistent with our framing of I(M;E) as a measure of action-outcome fusion with relevance to "empowerment." Our framing assumes that actions are encoded in terms of the sensory states they elicit — a well-supported idea from ideomotor theory (Fagioli et al., 2005; Hamilton et al., 2007; Hommel et al., 2001). On this view, the action performed during the "tile game" is encoded in terms of the sensory state brought about by the button-pressing motor command, which includes an active or inactive tile (i.e., the means, *M*). This would make *M*, as we define it, a component, or feature, of an action representation.

In our revised manuscript, we have made explicit how our definition of means relates to the concept of an action, thereby clarifying our theory's relevance to notions of empowerment and action-outcome fusion. Specifically, on lines 67-69 we describe our definition of M as: "...an echo of ideomotor theory, which proposes that actions are encoded in terms of the sensory states they elicit, rather than the motor commands that generate them."

Reviewer 2

R2.1. The revised manuscript addresses many of my comments and, from what I can tell, those of other reviewers. In particular, I appreciated the revamped data analysis, which I feel makes a clearer and more compelling case for the "computational theory of flow." Experiment 6 also provides additional evidence for a relationship between flow and mutual information across two real-life games. Overall, the authors convincingly demonstrate the core claim of the paper: that the mutual information between means and ends can be an important predictor of flow beyond alternative candidate measures based on the task's probabilistic structure (e.g. expected value) or self-report constructs (e.g. controllability). This is especially salient in experiments 3 and 5, where alternative measures are shown to make the wrong directional prediction about which of two tasks will generate more flow.

We thank Reviewer 2 for these positive comments and helpful feedback on our previous draft.

R2.2 In my opinion, a truly compelling empirical case for the theory would still require demonstrating that the specific non-linear functional relationship between task probabilities and flow predicted by mutual information holds up across a variety of contexts. This would determine whether mutual information itself drives flow, or whether mutual information merely correlates with the actual driver. Showing that mutual information and flow correlate across different games (as in experiment 6) is important, but ultimately does not address this more nuanced question. I will admit, however, that this is a heavy lift probably best left to future papers.

I also appreciated that the authors added further discussion of how the model might apply to real-life situations, such as reading a book or dancing the tango, but still came away feeling that the arguments presented here were a bit tenuous. It seems plausible, for instance, that someone might be uncertain about their ability to pull off a particular tango move, but I find it hard to imagine situations where someone is genuinely uncertain about their ability to continue reading a book, watching a movie, or listening to a gripping story. In its current form, therefore, I do not see how the theory can account for the very obvious but important fact that passive consumption of information seems to generate much of the flow people experience day to day. A similar point holds for environmental factors that do not change an activity's probabilistic contingencies. Perhaps the conclusion is simply that "means-ends fusion" is not the only determinant of flow; if this is the case, however, I think the paper would benefit from a more open and honest discussion of this point and its implications for our understanding of flow.

Finally, the authors also mention that "Most activities can be represented in multiple ways, making M and E perceiver-dependent," and indeed it strikes me that for many tasks means and ends can be redefined arbitrarily such that the mutual information between them is either large, small, or non-

existent. This is important because it means that the theory's predictions are entirely driven by assumptions about how people carve up the world into means and ends. As mentioned above, this is not at all obvious for many of the everyday things we do, meaning that different researchers can come to dramatically different conclusions about what the theory says. A theory is not complete unless there is a principled way of nailing down these researcher degrees of freedom. Is there a way of eliciting what people themselves define "means" to be? Hypothetically, is there a way to measure the probabilistic contingencies necessary to calculate the theory's prediction in real-world activities? Etc.

Reviewer 2's remaining critiques are three: (1) more work is needed to show that our theory of flow generalizes across activities, (2) $I(M;E)$ may not be the only determinant of flow, and (3) the perceiver-dependence of M and E limits the testability and applicability of our theory. We agree with all three critiques, and, per Reviewer 2's recommendation, have highlighted them in our revised discussion. On 365-377, we write:

"Two caveats deserve spotlighting. First, the present work does not suggest that $I(M;E)$ is the sole contributor to flow, nor does it suggest that $I(M;E)$ contributes to flow across all contexts. The informational theory of flow may yet be expanded by discoveries of additional inputs to the flow-generating process, and contracted by discoveries of contexts in which $I(M;E)$ fails to predict flow. A second caveat is that the quantity at the heart of our theory — $I(M;E)$ — is a function of variables whose properties are subjective. What M and E denote in a given task depends on how the person performing the task construes their means and end. On the one hand, the perceiver-dependence of M and E allows our theory to explain individual and situational differences in how much flow a particular activity elicits. On the other hand, it makes our theory difficult to apply in tasks with many possible means-end representations, a challenge we overcame by using tasks with clear means and ends. Expanding our theory to more ambiguous tasks hinges on the progress of ongoing research exploring how humans represent task structure. With better theories of how humans carve activities into means and ends, the informational theory of flow will become easier for researchers to falsify and for practitioners to apply."

In this passage, we emphasize that $I(M;E)$ may fail to predict flow in some situations. This is in line with Reviewer 2's suggestion to be more agnostic as to whether our theory can or cannot account for flow in all contexts. While highlighting the possibility that the relationship between flow and $I(M;E)$ applies to a subset of activities, this passage leaves open the possibility that it applies to all activities, and identifies this issue as an empirical question to be answered by future research. Accordingly, this passage implies that $I(M;E)$ may predict flow in the situations highlighted by Reviewer 2 as especially challenging for our theory, namely, activities like "reading a book, watching a movie, or listening to a gripping story." In what follows, we share why we think these activities may be more compatible with our theory than they appear.

Reviewer 2 notes that they "...find it hard to imagine situations where someone is genuinely uncertain about their ability to continue reading a book, watching a movie, or listening to a gripping story." Like Reviewer 2, we doubt that people are uncertain about

their *ability* to read books, watch movies, or listen to stories. Nonetheless, people may be uncertain about whether, or to what extent, they will actually do these things. Several popular treatments of goal-directed action and habit suggest that people learn and represent the probability of their own actions (e.g., Friston et al., 2015; Miller, Shenhav, & Ludvig, 2019; Solway & Botvinick, 2012). For instance, someone who reads every evening may learn a probability distribution over the number of pages they read per night. This knowledge can be used to construct a probability distribution over M in the book-reading context, one whose entropy could be substantial. The entropy of this distribution may be especially high when reading books in which chapter-endings are an unpredictable mix of cliffhangers that bait readers into continuing, and resolutions that serve as good stopping points — just the sort of books that tend to induce flow.

Regarding the example of watching a movie, we think this activity is less passive than it seems. Watching a movie (or any video) involves directing attention to informative and goal-relevant locations of a screen via eye movement (e.g., Itti & Baldi, 2006, 2009; van Zoest et al., 2004). Thus, in the movie-viewing context, uncertainty about M may correspond to uncertainty about future fixation points, which, in many viewing contexts, may be quite high.

Finally, books and movies both invite viewers to simulate the actions and experiences of the characters, so some of the immersion they elicit may be attributable to readers vividly imagining high- $I(M;E)$ experiences. Consider, for instance, Agatha Christie's most famous detective, Hercule Poirot. Poirot begins each novel highly uncertain of whom he'll reveal as the murderer, and completely certain that whomever he reveals will be brought to justice (the killer is invariably hauled away immediately after the big reveal with the exception of *Murder on the Orient Express*). Thus, for Poirot, there is high mutual information between whom he reveals as the murderer, M , and whom is punished, E . This may induce flow in readers taking Poirot's perspective.

We present these hypotheses merely to suggest that the situations described by Reviewer 2 — reading, watching movies, etc. — may be less problematic for our theory than they appear. As we noted above, Reviewer 2's prediction that flow is unrelated to $I(M;E)$ in these activities may prove correct. Addressing this open question will be a key focus of our future work on this topic.

R2.3. The above points of critique aside, I am enthusiastic about the creative and provocative claim at the core of this paper. Personally, I think the paper in its current form makes a significant scientific contribution worth sharing with other researchers. I also think, however, that a paper which seeks to revolutionize our understanding of a concept is most scientifically useful if it lays out exactly how it can be applied to situations beyond its author's own experiments. In particular, the present paper would be greatly strengthened if it fully "closed the theory" by clarifying exactly how it could be operationalized in non-lab settings (particularly, by addressing the fraction problem of defining and measuring an activity's "means"). I would also recommend space be given for a more direct discussion of how (or whether) the theory can account for stylized features of flow that do not neatly fit into the a means-end framework, e.g. that passive information consumption can generate flow.

We thank Reviewer 2 for their helpful feedback, which we have addressed with a thorough discussion of limitations and future directions (see R2.2).

Reviewer 3

R3.1. The authors have addressed all my concerns and comments. I recommend publication.

REVIEWERS' COMMENTS

Reviewer #1 (Remarks to the Author):

The authors have provided a satisfactory reply to my one remaining concern and I am pleased to recommend publication.

Reviewer #2 (Remarks to the Author):

The authors inserted a major disclaimer that acknowledges the framework's limitations and ambiguities. On the whole, I think the paper now does a balanced job of presenting a provocative hypothesis while making clear that it is not fully worked out and needs further study. I would recommend publication at this point